# Video Prediction Models as Rewards
# for Reinforcement Learning

**Alejandro Escontrela**[†][*]     **Ademi Adeniji**[†]     **Wilson Yan**[†]

**Ajay Jain**     **Xue Bin Peng**     **Ken Goldberg**

**Youngwoon Lee**     **Danijar Hafner**     **Pieter Abbeel**

University of California, Berkeley

[†]Equal contribution

## Abstract

Specifying reward signals that allow agents to learn complex behaviors is a long-standing challenge in reinforcement learning. A promising approach is to extract preferences for behaviors from unlabeled videos, which are widely available on the internet. We present Video Prediction Rewards (VIPER), an algorithm that leverages pretrained video prediction models as action-free reward signals for reinforcement learning. Specifically, we first train an autoregressive transformer on expert videos and then use the video prediction likelihoods as reward signals for a reinforcement learning agent. VIPER enables expert-level control without programmatic task rewards across a wide range of DMC, Atari, and RLBench tasks. Moreover, generalization of the video prediction model allows us to derive rewards for an out-of-distribution environment where no expert data is available, enabling cross-embodiment generalization for tabletop manipulation. We see our work as starting point for scalable reward specification from unlabeled videos that will benefit from the rapid advances in generative modeling. Source code and datasets are available on the project website: https://escontrela.me/viper

## 1 Introduction

Manually designing a reward function is laborious and often leads to undesirable outcomes [32]. This is a major bottleneck for developing general decision making agents with reinforcement learning (RL). A more scalable approach is to learn complex behaviors from *videos*, which can be easily acquired at scale for many applications (e.g., Youtube videos).

Previous approaches that learn behaviors from videos reward the similarity between the agent's current observation and the expert data distribution [37, 38, 30, 43]. Since their rewards only condition on the current observation, they cannot capture temporally meaningful behaviors. Moreover, the approaches with adversarial training schemes [30, 43] often result in mode collapse, which hinders generalization.

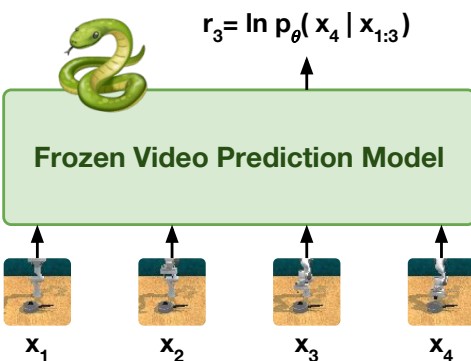

Figure 1: VIPER uses the next-token likelihoods of a frozen video prediction model as a general reward function for various tasks.

37th Conference on Neural Information Processing Systems (NeurIPS 2023).

[*]Corresponding author: escontrela@berkeley.edu

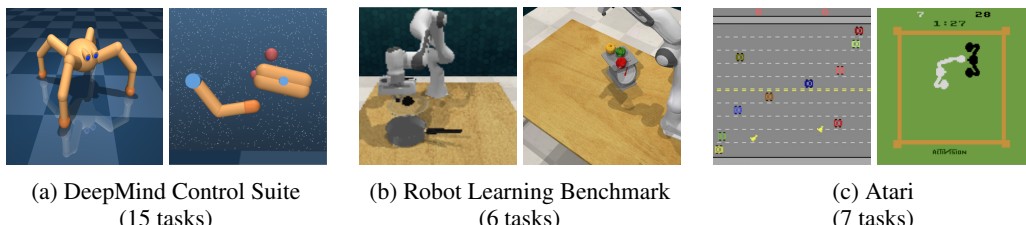

|(a) DeepMind Control Suite|(b) Robot Learning Benchmark|(c) Atari|
|(15 tasks)|(6 tasks)|(7 tasks)|

Figure 2: VIPER achieves expert-level control directly from pixels without access to ground truth rewards or expert actions on 28 reinforcement learning benchmark tasks.

Other works fill in actions for the (action-free) videos using an inverse dynamics model [42, 8]. Dai et al. [8] leverage recent advances in generative modeling to capture multi-modal and temporally-coherent behaviors from large-scale video data. However, this multi-stage approach requires performing expensive video model rollouts to then label actions with a learned inverse dynamics model.

In this paper, we propose using Video Prediction Rewards (VIPER) for reinforcement learning. VIPER first learns a video prediction model from expert videos. We then train an agent using reinforcement learning to maximize the log-likelihood of agent trajectories estimated by the video prediction model, as illustrated in Figure 1. Directly leveraging the video model's likelihoods as a reward signal encourages the agent to match the video model's trajectory distribution. Additionally, rewards specified by video models inherently measure the temporal consistency of behavior, unlike observation-level rewards. Further, evaluating likelihoods is significantly faster than performing video model rollouts, enabling faster training times and more interactions with the environment.

We summarize the three key contributions of this paper as follows:

- We present VIPER: a novel, scalable reward specification algorithm which leverages rapid improvements in generative modeling to provide RL agents with rewards from unlabeled videos.
- We perform an extensive evaluation, and show that VIPER can achieve expert-level control **without task rewards** on 15 DMC tasks [44], 6 RLBench tasks [20], and 7 Atari tasks [3] (see examples in Figure 2 and Appendix A.5).
- We demonstrate that VIPER generalizes to different environments for which no training data was provided, enabling cross-embodiment generalization for tabletop manipulation.

Along the way, we discuss important implementation details that improve the robustness of VIPER.

## 2 Related Work

Learning from observations is an active research area which has led to many advances in imitation learning and inverse reinforcement learning [27, 1, 49]. This line of research is motivated by the fact that learning policies from expert videos is a scalable way of learning a wide variety of complex tasks, which does not require access to ground truth rewards, expert actions, or interaction with the expert. Scalable solutions to this problem would enable learning policies using the vast quantity of videos on the internet. Enabling policy learning from expert videos can be largely categorized into two approaches: (1) Behavioral cloning [31] on expert videos labelled with predicted actions and (2) reinforcement learning with a reward function learned from expert videos.

**Labelling videos with predicted actions**  An intuitive way of leveraging action-free expert videos is to guess which action leads to each transition in expert videos, then to mimic the predicted actions. Labelling videos with actions can be done using an inverse dynamics model, $p(a|s, s')$, which models an action distribution given the current and next observations. An inverse dynamics model can be learned from environment interactions [28, 42, 29] or a curated action-labelled dataset [2, 34]. Offline reinforcement learning [24] can be also used instead of behavioral cloning for more efficient use of video data with predicted actions [34]. However, the performance of this approach heavily depends on the quality of action labels predicted by an inverse dynamics model and the quantity and diversity of training data.

**Reinforcement learning with videos**   Leveraging data from online interaction can further improve policies trained using unlabelled video data. To guide policy learning, many approaches have learned a reward function from videos by estimating the progress of tasks [37, 38, 25, 23] or the divergence between expert and agent trajectories [43, 30]. Adversarial imitation learning approaches [43, 30] learn a discriminator that discriminates transitions from the expert data and the rollouts of the current policy. Training a policy to maximize the discriminator error leads to similar expert and policy behaviors. However, the discriminator is prone to mode collapse, as it often finds spurious associations between task-irrelevant features and expert/agent labels [50, 19], which requires a variety of techniques to stabilize the adversarial training process [7]. In contrast, VIPER directly models the expert video distribution using recent generative modeling techniques [10, 47], which offers stable training and strong generalization.

**Using video models as policies**   Recently, UniPi [8] uses advances in text-conditioned video generation models [33] to plan a trajectory. Once the future video trajectory is generated, UniPi executes the plan by inferring low-level controls using a learned inverse dynamics model. Instead of using slow video generations for planning, VIPER uses video prediction likelihoods to guide online learning of a policy.

# 3   Video Prediction Rewards

In this section, we propose Video Prediction Rewards (VIPER), which learns complex behaviors by leveraging the log-likelihoods of pre-trained video prediction models for reward specification. Our method does not require any ground truth rewards or action annotations, and only requires videos of desired agent behaviors. VIPER implements rewards as part of the environment, and can be paired with any

---

**Algorithm 1:** VIPER

Train video prediction model $p_\theta$ on expert videos.
**while** *not converged* **do**
  Choose action: $a_t \sim \pi(x_t)$
  Step environment: $x_{t+1} \leftarrow \mathrm{env}(a_t)$
  Fill in reward: $r_t \leftarrow \ln p_\theta(x_{t+1} \mid x_{t-k:t}) + \beta r_t^{\mathrm{expl}}$
  Add transition $(x_t, a_t, r_t, x_{t+1})$ to replay buffer.
  Train $\pi$ from replay buffer using any RL algorithm.

---

RL algorithm. We overview the key components of our method below.

## 3.1   Video Modeling

Likelihood-based video models are a popular paradigm of generative models trained to model the data distribution by maximizing an exact or lower bound on the log-likelihood of the data. These models have demonstrated their ability to fit highly multi-modal distributions and produce samples depicting complex dynamics, motion, and behaviors [40, 18, 17, 45].

Our method can integrate any video model that supports computing likelihoods over the joint distribution factorized in the following form:

$$\log p(x_{1:T}) = \sum_{t=1}^{T} \log p(x_t \mid x_{1:t-1}), \tag{1}$$

where $x_{1:T}$ is the full video consisting of $T$ frames, $x_1, \ldots, x_T$. When using video models with limited context length $k$, it is common to approximate likelihoods of an entire video sequence with its subsequences of length $k$ as follows:

$$\log p(x_{1:T}) \approx \sum_{t=1}^{T} \log p(x_t \mid x_{\max(1,t-k):t-1}). \tag{2}$$

In this paper, we use an autoregressive transformer model based on VideoGPT [47, 36] as our video generation model. We first train a VQ-GAN [10] to encode individual frames $x_t$ into discrete codes $z_t$. Next, we learn an autoregressive transformer to model the distribution of codes $z$ through the following maximum likelihood objective:

$$\max_\theta \sum_{t=1}^{T} \sum_{i=1}^{Z} \log p_\theta(z_t^i \mid z_t^{1:i-1}, z_{1:t-1}), \tag{3}$$

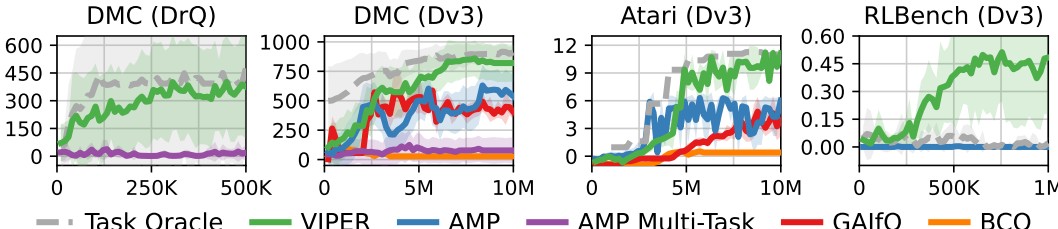

Figure 3: Aggregate results across 15 DMC tasks, 7 Atari games, and 6 RLBench tasks. DMC results are provided for DrQ and DreamerV3 (Dv3) RL agents. Atari and RLBench results are reported for DreamerV3. Atari scores are computed using Human-Normalized Mean.

where $z_t^i$ is the $i$-th code of the $t$-th frame, and $Z$ is the total number of codes per frame. Computing the exact conditional likelihoods $p_\theta(x_t \mid x_{1:t-1})$ is intractable, as it requires marginalizing over all possible combinations of codes. Instead, we use the conditional likelihoods over the latent codes $p_\theta(z_t \mid z_{1:t-1})$ as an approximation. In our experiments, we show that these likelihoods are sufficient to capture the underlying dynamics of the videos.

Note that our choice of video model does not preclude the use of other video generation models, such as MaskGIT-based models [48, 45, 13] or diffusion models [18, 40]. However, we opted for an autoregressive model due to the favorable properties of being able to model complex distributions while retaining fast likelihood computation. Video model comparisons are performed in Section 4.5.

### 3.2 Reward Formulation

Given a pretrained video model, VIPER proposes an intuitive reward that maximizes the conditional log-likelihoods for each transition $(x_t, a_t, x_{t+1})$ observed by the agent:

$$r_t^{\text{VIPER}} \doteq \ln p_\theta(x_{t+1} \mid x_{1:t}). \tag{4}$$

This reward incentivizes the agent to find the most likely trajectory under the expert video distribution as modeled by the video model. However, the most probable sequence does not necessarily capture the distribution of behaviors we want the agent to learn.

For example, when flipping a weighted coin with $p(\text{heads} = 0.6)$ 1000 times, *typical sequences* will count roughly 600 heads and 400 tails, in contrast to the most probable sequence of 1000 heads that will basically never be seen in practice [39]. Similarly, the most likely image under a density model trained on MNIST images is often the image of only background without a digit, despite this never occurring in the dataset [26]. In the reinforcement learning setting, an additional issue is that solely optimizing a dense reward such as $r_t^{\text{VIPER}}$ can lead to early convergence to local optima.

To overcome these challenges, we take the more principled approach of matching the agent's trajectory distribution $q(x_{1:T})$ to the sequence distribution $p_\theta(x_{1:T})$ of the video model by minimizing the KL-divergence between the two distributions [41, 15]:

$$\text{KL}\big[q(x_{1:T}) \,\big\|\, p(x_{1:T})\big] = \underbrace{\text{E}_q\big[-\ln p_\theta(x_{1:T})\big]}_{\text{cross-entropy}} - \underbrace{\text{H}\big[q(x_{1:T})\big]}_{\text{entropy}}$$

$$= -\sum_{t=1}^{T}\left[ \text{E}_q\big[\underbrace{\ln p_\theta(x_{t+1} \mid x_{1:t})}_{r_t^{\text{VIPER}}}\big] + \underbrace{\text{H}\big[q(x_{t+1} \mid x_{1:t})\big]}_{\text{exploration term}}\right], \tag{5}$$

The KL objective shows that for the agent to match the video distribution under the video prediction model, it has to not only maximize the VIPER reward but also balance this reward while maintaining high entropy over its input sequences [21]. In the reinforcement learning literature, the entropy bonus over input trajectories corresponds to an exploration term that encourages the agent to explore and inhabit a diverse distribution of sequences within the regions of high probability under the video prediction model. This results in the final reward function that the agent maximizes:

$$r_t^{\text{KL}} \doteq r_t^{\text{VIPER}} + \beta\, r_t^{\text{expl}}, \tag{6}$$

where $\beta$ determines the amount of exploration. To efficiently compute the likelihood reward, we approximate the context frames with a sliding window as discussed in Equation 2:

$$r_t^{\text{VIPER}} \approx \ln p_\theta(x_{t+1} \mid x_{\max(1,t-k):t}). \tag{7}$$

Figure 1 shows the process of computing rewards using log probabilities under the video prediction model. VIPER is agnostic to the choice of exploration reward and in this paper we opt for Plan2Explore [35] and RND [5].

## 3.3 Data Curation

In this work, we explore whether VIPER provides adequate reward signal for learning low-level control. We utilize the video model likelihoods provided by an autoregressive video prediction model pre-trained on data from a wide variety of environments. We curate data by collecting expert video trajectories from task oracles and motion planning algorithms with access to state information. Fine-tuning large text-conditioned video models [33, 45, 40] on expert videos would likely lead to improved generalization performance beyond the curated dataset, and would make for an interesting future research direction. We explore the favorable generalization capabilities of video models trained on small datasets, and explore how this leads to more general reward functions in Section 4.3.

## 4 Experiments

We evaluate VIPER on 28 different tasks across the three domains shown in Figure 2. We utilize 15 tasks from the DeepMind Control (DMC) suite [44], 7 tasks from the Atari Gym suite [4], and 6 tasks from the Robot Learning Benchmark (RLBench) [20]. We compare VIPER agents to variants of Adversarial Motion Priors (AMP) [30], which uses adversarial training to learn behaviors from reward-free and action-free expert data. All agents are trained using raw pixel observations from the environment, with no access to state information or task rewards. In our experiments, we aim to answer the following questions:

1. Does VIPER provide an adequate learning signal for solving a variety of tasks? (Section 4.2)
2. Do video models trained on many different tasks still provide useful rewards? (Section 4.2)
3. Can the rewards generalize to novel scenarios where no expert data is available? (Section 4.3)
4. How does the video model data quality and quantity affect the learned rewards? (Section 4.4)
5. What implementation details matter when using video model likelihoods as rewards? (Section 4.5)

## 4.1 Video Model Training Details

**Training data** To collect expert videos in DMC and Atari tasks, **Task Oracle** RL agents are trained until convergence with access to ground truth state information and task rewards. After training, videos of the top $k$ episodes out of 1000 episode rollouts for each task are sampled as expert videos, where $k = 50$ and 100 for DMC and Atari, respectively. For RLBench, a sampling-based motion planning algorithm with access to full state information is used to gather 100 demos for each task, where selected tasks range from "easy" to "medium".

**Video model training** We train a *single* autoregressive video model for each suite of tasks. Example images for each suite are shown in Appendix A.5. For DMC, we train a single VQ-GAN across all tasks that encodes $64 \times 64$ images to $8 \times 8$ discrete codes. Similarly for RLBench and Atari, we train VQ-GANs across all tasks within each domain, but encode $64 \times 64$ images to $16 \times 16$ discrete codes. In practice, the level of VQ compression depends on the visual complexity of the environment – more texturally simple environments (e.g., DMC) allow for higher levels of spatial compression. We follow the original VQ-GAN architecture [10], consisting of a



Figure 4: Video model rollouts for 3 different evaluation environments.

CNN encoder and decoder. We train VQ-GAN with a batch size of 128 and learning rate $10^{-4}$ for 200k iterations.

To train our video models, we encode each frame of the trajectory and stack the codes temporally to form an encoded 3D video tensor. Similar to VideoGPT [47], the encoded VQ codes for all video

frames are jointly modeled using an autoregressive transformer in raster scan order. For DMC and Atari, we train on 16 frames at a time with a one-hot label for task conditioning. For RLBench, we train both single-task and multi-task video models on 4 frames with frame skip 4. We add the one-hot task label for the multi-task model used in our cross-embodiment generalization experiments in Section 4.3. We perform task conditioning identical to the class conditioning mechanism in VideoGPT, with learned gains and biases specific to each task ID for each normalization layer.

Figure 4 shows example video model rollouts for each domain. In general, our video models are able to accurately capture the dynamics of each environment to produce probable futures. Further details on model architecture and hyperparameters can be found in Appendix A.2. All models are trained on TPUv3-8 instances which are approximately similar to 4 Nvidia V100 GPUs.

## 4.2 Video Model Likelihoods as Rewards for Reinforcement Learning

To learn behaviors from expert videos, we provide reward signals using Equation 6 in VIPER and the discriminator error in AMP. Both VIPER and AMP are agnostic to the choice of RL algorithm; but, in this paper, we evaluate our approach and baselines with two popular RL algorithms: DrQ [22] and DreamerV3 [16]. We use Random Network Distillation [5] (model-free) as the exploration objective for DrQ, and Plan2Explore [35] (model-based) for DreamerV3. Hyperparameters for each choice of RL algorithm are shown in Appendix A.3. We compare VIPER to two variants of the AMP algorithm: the single-task variant where only expert videos for the specific task are provided to the agent, and the multi-task case where expert videos for all tasks are provided to the agent.

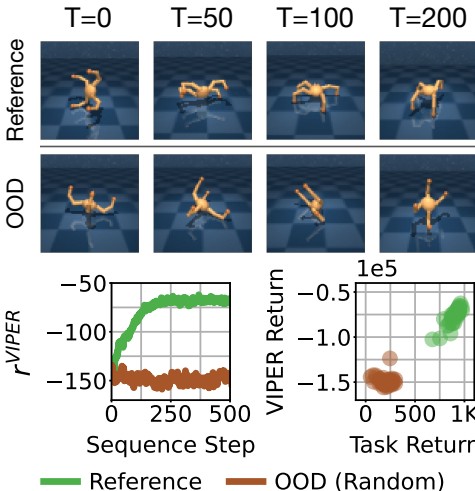

Figure 5: VIPER incentivizes the agent to maximize trajectory likelihood under the video model. As such, it provides high rewards for reference (expert) sequences, and low rewards for unlikely behaviors. Mean rewards $r^{\mathrm{VIPER}}$ and returns are computed across 40 trajectories.

As outlined in Algorithm 1, we compute Equation 6 for every environment step to label the transition with a reward before adding it to the replay buffer for policy optimization. In practice, batching Equation 6 across multiple environments and leveraging parallel likelihood evaluation leads to only a small decrease in training speed. DMC agents were trained using 1 Nvidia V100 GPU, while Atari and RLBench agents were trained using 1 Nvidia A100 GPU.

We first verify whether VIPER provides meaningful rewards aligned with the ground truth task rewards. Figure 5 visualizes the ground truth task rewards and our log-likelihood rewards (Equation 7) for a reference (expert) trajectory and a random out-of-distribution trajectory. In the reward curves on the left, VIPER starts to predict high rewards for the expert transitions once the transitions becomes distinguishable from a random trajectory, while it consistently outputs low rewards for out-of-distribution transitions. The return plot on the right clearly shows positive correlation between returns of the ground truth reward and our proposed reward.

Then, we measure the task rewards when training RL agents with predicted rewards from VIPER and baselines and report the aggregated results for each suite and algorithm in Figure 3. The full results can be found in Appendix A.4.

In DMC, VIPER achieves near expert-level performance from pixels with our video prediction rewards alone. Although VIPER slightly underperforms Task Oracle, this is surprising as the Task Oracle uses *full state information* along with *dense task rewards*. VIPER outperforms both variants of AMP. Worth noting is the drastic performance difference between the single-task and multi-task AMP algorithms. This performance gap can possibly be attributed to mode collapse, whereby the discriminator classifies all frames for the current task as fake samples, leading to an uninformative reward signal for the agent. Likelihood-based video models, such as VIPER, are less susceptible to mode collapse than adversarial methods.

In Atari, VIPER approaches the performance of the Task Oracle trained with the original sparse task reward, and outperforms the AMP baseline. Shown in Figure 6, we found that masking the scoreboard in each Atari environment when training the video model improved downstream RL performance.

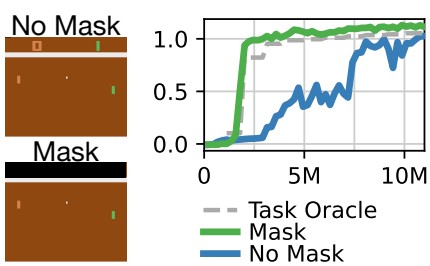

Figure 6: RL training curves on Atari Pong when using VIPER trained with or without masking the scoreboard.

Since the video model learns to predict all aspects of the data distribution, including the scoreboard, it tends to provide noisier reward signal during policy training when the agent encounters out-of-distributions scores not seen by the video model. For example, expert demos for Pong only contain videos where the player scores and the opponent's score is always zero. During policy training, we observe that the Pong agent tends to exhibit more erratic behaviors as soon as the opponent scores at least one point, whereas when masking the scoreboard, learned policies are generally more stable. These results suggest the potential benefits of finetuning large video models on expert demos to learn more generalizable priors, as opposed to training from scratch.

For RLBench, VIPER outperforms the Task Oracle because RLBench tasks provide very sparse rewards after long sequences of actions, which pose a challenging objective for RL agents. VIPER instead provides a dense reward extracted from the expert videos, which helps learn these challenging tasks. When training the video model, we found it beneficial to train at a reduced frame rate, accomplished by subsampling video sequences by a factor of 4. Otherwise, we observed the video model would assign high likelihoods to stationary trajectories, resulting in learned policies rarely moving and interacting with the scene. We hypothesize that this may be partially due to the high control frequency of the environment, along with the initial slow acceleration of the robot arm in demonstrations, resulting in very little movement between adjacent frames. When calculating likelihoods for reward computation, we similarly input observations strided by time, e.g., $p(x_t \mid x_{t-4}, x_{t-8}, \dots)$.

### 4.3 Generalization of Video Prediction Rewards

Prior works in the space of large, pretrained generative models have shown a powerful ability to generalize beyond the original training data, demonstrated by their ability to produce novel generations (e.g., unseen text captions for image or video generation [33, 45, 40]) as well as learn more generalizable models from limited finetuning data [8, 46]. Both capabilities provide promising directions for extending large video generation models to VIPER, where we can leverage text-video models to capture priors specified by text commands, or finetune on a small set of demonstrations to learn a task-specific prior that better generalizes to novel states.

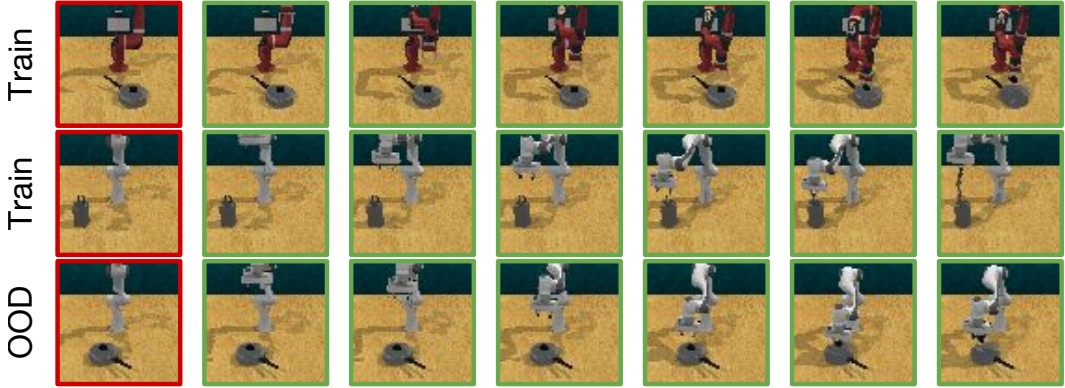

Figure 7: Sampled video predictions for in distribution reference videos (Train) and an OOD arm/task combination (OOD). The video model displays cross-embodiment generalization to arm/task combination not observed in the training data. Video model generalization can enable specifying new tasks where no reference data is available.

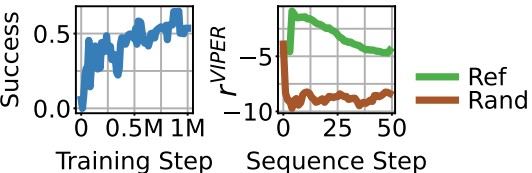

In this section, we seek to understand how this generalization can be used to learn more general reward functions. We train a model on two datasets of different robot arms, and evaluate the *cross-embodiment generalization* capabilities of the model. Specifically, we gather demonstrations for 23 tasks on the Rethink Robotics Sawyer Arm, and demonstrations for 30 tasks on the Franka Panda robotic arm, where only 20 tasks are overlapping between arms. We then train a task-conditioned autoregressive video model on these demonstration videos and evaluate the video model by querying unseen arm/task combinations, where a single initial frame is used for open loop predictions.

Figure 8: (Left) Training curve for RL agent trained with VIPER on OOD task. (Right) Task-conditional likelihood for reference and random trajectory for an OOD task.

Sample video model rollouts for in distribution training tasks and an OOD arm/task combination are shown in Figure 7. Even though the video model was not directly trained on demonstrations of the Franka Panda arm to solve the saucepan task in RLBench, it is able to generate reasonable trajectories for the arm and task combination. Figure 8 further validates this observation by assigning higher likelihood to expert videos (Ref) compared to random robot trajectories (Rand). We observe that these generalization capabilities also extend to downstream RL, where we use our trained video model with VIPER to learn a policy for the Franka Robot arm to solve an OOD task without requiring demos for that specific task and arm combination. Figure 21 further extends this analysis to include more in distribution and out of distribution tasks. These results demonstrate a promising direction for future work in applying VIPER to larger scale video models that will be able to better generalize and learn desired behaviors only through a few demonstrations.

## 4.4 Impact of Data Quality and Quantity

Learning from sub-optimal data is an important feature of learned reward models, as large-scale video data may often contain suboptimal solutions to a given task. As such, we evaluate VIPER's ability to learn rewards from sub-optimal data.

We train video models with suboptimal (good) data, which has 50-75% returns of the expert data. In Figure 9, VIPER learns the suboptimal behaviors provided in the suboptimal video data. This suboptimal VIPER can be still useful if combined with a sparse task reward, comparable to an agent learned from dense task rewards.

Additionally, we evaluate how VIPER performs under different video dataset sizes. As shown in Figure 10, VIPER can learn a meaningful reward function only with one expert trajectory, although adding more videos quickly improves the performance of VIPER, with diminishing returns after 50 trajectories.

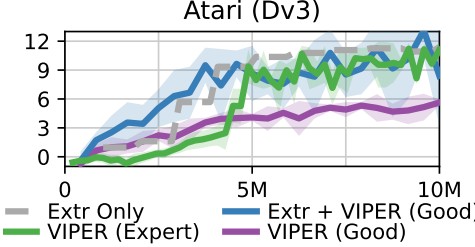

Figure 9: Atari performance with VIPER models trained on suboptimal data.

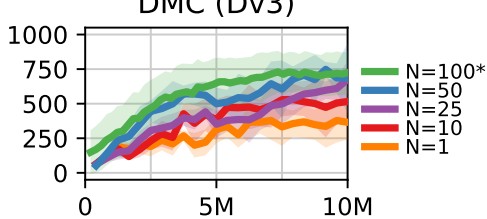

Figure 10: DMC performance with VIPER trained on different dataset sizes, where N is the number of expert trajectories. *Original dataset size.

## 4.5 Ablation Studies

In this section, we study the contributions of various design decisions when using video prediction models as rewards: (1) how to weight the exploration objective, (2) which video model to use, and (3) what context length to use. Ablation studies are performed across DMC tasks.

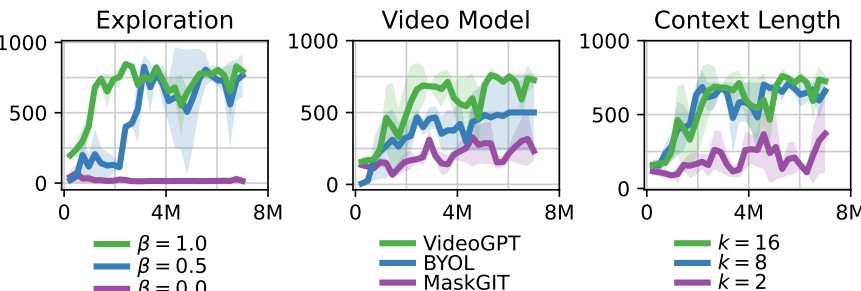

Figure 11: Effect of exploration reward term, video model choice, and context length on downstream RL performance. An equally weighted exploration reward term and longer video model context leads to improved performance. MaskGIT substantially underperforms VideoGPT as a choice of video model. The BYOL model performs moderately due to the deterministic architecture not properly handling multi-modality.

**Exploration objective**    As discussed in subsection 3.2, an exploration objective may help the RL agent learn a distribution of behaviors that closely matches the distribution of the original data and generalizes, while preventing locally optimal solutions. In Figure 11, we ablate the $\beta$ parameter introduced in Equation 6 using a VideoGPT-like model as the VIPER backbone. $\beta = 0$ corresponds to no exploration objective, whereas $\beta = 1$ signifies equal weighting between $r^{\text{VIPER}}$ and $r^{\text{expl}}$. We ablate the exploration objective using the Plan2Explore [35] reward, which provides the agent with a reward proportional to the disagreement between an ensemble of one-step dynamics models. Using no exploration objective causes the policy's behavior to collapse, while increasing the weight of the exploration objective leads to improved performance.

**Video model**    Although our experiments focus on using an autoregressive VideoGPT-like model to compute likelihoods, VIPER generally allows for any video model that supports computing conditional likelihoods or implicit densities.

Figure 11 shows additional ablations replacing our VideoGPT model with a similarly sized MaskGIT [6] model, where frames are modeled with MaskGIT over space, and autoregressive over time. MaskGIT performs substantially worse than the VideoGPT model, which is possibly due to noisier likelihoods from parallel likelihood computation. In addition, while VideoGPT only requires 1 forward pass to compute likelihood, MaskGIT requires as many as 8 forward passes on the sequence of tokens for a frame, resulting in an approximately $8\times$ slowdown in reward computation, and $2.5\times$ in overall RL training.

Finally, we evaluate the performance of a video model that computes implicit densities using a negative distance metric between online predictions and target encodings with a recurrent Bootstrap Your Own Latent (BYOL) architecture [12, 11]. We refer the reader to Appendix A.1 for more details about the implementation. While BYOL outperforms MaskGIT in Figure 11, its deterministic recurrent architecture is unable to predict accurate embeddings more than a few steps into the future. This limits the ability of the learned reward function to capture temporal dynamics. We observe that agents trained with BYOL often end up in a stationary pose, which achieves high reward from the BYOL model.

**Context length**    The context length $k$ of the video model is an important choice in determining how much history to incorporate into the conditional likelihood. At the limit where $k = 0$, the reward is the unconditional likelihood over each frame. Such a choice would lead to stationary behavior on many tasks. Figure 11 shows that increasing the context length can help improve performance when leveraging a VideoGPT-based model for downstream RL tasks, subject to diminishing returns. We hypothesize that a longer context length may help for long-horizon tasks.

# 5   Conclusion

This paper presents Video Prediction Rewards (VIPER), a general algorithm that enables agents to learn behaviors from videos of experts. To achieve this, we leverage a simple pre-trained autoregressive video model to provide rewards for a reinforcement learning agent. These rewards are parameterized as the conditional probabilities of a particular frame given a context past of frames. In addition, we include an entropy maximization objective to ensure that the agent learns diverse behaviors which match the video model's trajectory distribution. VIPER succeeds across 3 benchmarks and 28 tasks, including the DeepMind Control Suite, Atari, and the Reinforcement Learning Benchmark. We find that simple data augmentation techniques can dramatically improve the effectiveness of VIPER. We also show that VIPER generalizes to out-of-distribution tasks for which no demonstrations were provided. Moreover, VIPER can learn reward functions for a wide variety of tasks using a single video model, outperforming AMP [30], which suffers from mode collapse as the diversity of the expert videos increases.

Limitations of our work include that VIPER is trained using in-domain data of expert agents, which is not a readily-available data source in the real world. Additionally, video prediction rewards trained on stochastic data may lead the agent to prefer states that minimize the video model's uncertainty, which may lead to sub-optimal behaviors. This challenge may be present for cases where either the environment is stochastic or the demonstrator provides noisy demonstrations. Moreover, selecting the right trade-off between VQCode size and context length has a significant impact on the quality of the learned rewards, with small VQCodes failing to model important components of the environment (e.g., the ball in Atari Breakout) and short context lengths leading to myopic behaviors which results in poor performance. Exploring alternative video model architectures for VIPER would likely be a fruitful research direction.

To improve the generalization capabilities of VIPER, larger pre-trained video models are necessary. Future work will explore how fine-tuning or human preferences can be used to improve video prediction rewards. Using text-to-video models remain another interesting direction for future work in extracting text-conditioned task-specific priors from a pretrained video model. In addition, extending this line of work to leverage Video Diffusion Models [18] as video prediction rewards may lead to interesting outcomes.

## Acknowledgments and Disclosure of Funding

This work was supported in part by an NSF Fellowship, NSF NRI #2024675, ONR MURI N00014-22-1-2773, Komatsu, and the Vanier Canada Graduate Scholarship. We also thank Google TPU Research Cloud and Cirrascale (https://cirrascale.com) for providing compute resources.

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

## A  Appendix

### A.1  Bootstrap Your Own Latent Details

**Model Architecture.** The BYOL model used in this work derives from the recurrent BYOL-Explore architecture proposed in [12], with a few modifications. Namely Guo et al. propose BYOL-Explore, a multi-step predictive latent world model. The BYOL-Explore architecture trains an online network using targets generated by an exponential moving average target network on future observations. Observations $o_t$ are first encoded into an observation representation $f_\theta(o_t)$. Online predictions are computed by then encoding the observation representation using a closed-loop RNN $h_\theta^c(f_\theta(o_t))$. Guo et al. also pass actions into the closed loop RNN, but we wish to learn an action-free reward mechanism that can be trained solely from videos. At each timestep, the carry $b_t$ from the closed loop RNN is then fed into an open-loop RNN $h_\theta^o$ which predicts open-loop representations $K$ steps into the future: $(b_{t,k} \in \mathcal{R}^\mathcal{M})_{k=1}^{K-1}$, where $b_{t,k} = h_\theta^o(b_{t,k-1})$. The original BYOL-Explore architecture feeds actions $a_{t+k+1}$ as inputs to the open loop cell $h_\theta^o$ as well, but we opt to omit these inputs for the same reason outlined above.

Given the deterministic RNN architecture used to predict online targets, omitting the action conditioning may lead to poor performance in multi-modal or stochastic environments. A more principled approach would utilize a probabilistic recurrent state space model [9, 14] to account for the multi-modality of futue states. We leave this approach to future work.

Finally, a predictor head $g_\theta(b_{t,k})$ outputs online predictions. We refer the reader to [12] for a figure of the outlined architecture.

The target network is an observation encoder $f_\phi$ whose parameters are an exponential moving average of $f_\theta$. The loss function for the online network is then defined as:

$$\mathcal{L}_{\text{BYOL-Explore}}(\theta, t, k) = \left\| \frac{g_\theta(b_{t,k})}{\|g_\theta(b_{t,k})\|_2} - \mathsf{sg}\left( \frac{f_\phi(o_{t+k})}{\|f_\phi(o_{t+k})\|_2} \right) \right\|_2^2,$$

$$\mathcal{L}_{\text{BYOL-Explore}}(\theta) = \frac{1}{B(T-1)} \sum_{t=0}^{T-2} \frac{1}{K(t)} \sum_{k=1}^{K(t)} \mathcal{L}_{\text{BYOL-Explore}}(\theta, t, k),$$

where $K(t) = \min(K, T-1-t)$ is the valid open-loop horizon for a trajectory of length $T$ and $\mathsf{sg}$ is the stop-gradient operator.

**Computing Rewards.** The uncertainty associated with the transition $(o_t, a_t, o_{t+1})$ is the sum of the corresponding prediction losses:

$$\ell_t = \sum_{p+q=t+1} \mathcal{L}_{\text{BYOL-Explore}}(\theta, j, p, q),$$

where $0 \leq p \leq T-2$, $1 \leq q \leq K$ and $0 \leq t \leq T-2$. This accumulates all the losses corresponding to the latent dynamics model uncertainties relative to the observation $o_{t+1}$. For our purposes, we calculate rewards as $r_t = -\ell_t$, which is equivalent to negating the reward used in the original BYOL-Explore formulation, and can be interpreted as negating the exploration objective to ensure that the agent stays close to the trajectory distribution found in the original dataset.

## A.2 Video Model Hyperparameters

Table 1: Hyperparameters and training details for all VQ-GAN models

| | DMC | Atari | RLBench (single + multi task) |
|---|---|---|---|
| Input size | $64 \times 64 \times 3$ | $64 \times 64 \times 3$ | $64 \times 64 \times 3$ |
| Latent size | $8 \times 8$ | $16 \times 16$ | $16 \times 16$ |
| $\beta$ (commitment loss coefficient) | 0.25 | 0.25 | 0.25 |
| Batch size | 128 | 128 | 128 |
| Learning rate | $10^{-4}$ | $10^{-4}$ | $10^{-4}$ |
| Learning rate schedule | constant | constant | constant |
| Training steps | 200k | 200k | 200k |
| Base channels | 128 | 128 | 128 |
| Ch mult | [1, 1, 2, 2] | [1, 2, 2] | [1, 2, 2] |
| Num res blocks | 1 | 1 | 2 |
| Codebook size | 1024 | 1024 | 1024 |
| Codebook dimension | 64 | 64 | 64 |
| Perceptual loss weight | 0.1 | 0.1 | 0.1 |
| Disc base features | 32 | 32 | 32 |
| Disc gradient penalty weight | $10^8$ | $10^8$ | $10^8$ |
| Disc max hidden feature size | 512 | 512 | 512 |
| Disc mbstd group size | 4 | 4 | 4 |
| Disc mbstd num features | 1 | 1 | 1 |
| Disc loss weight | 0.1 | 0.1 | 0.1 |

Table 2: Hyperparameters and training details for all VideoGPT and MaskGit models

| | DMC | Atari | RLBench (single task) | RLBench (multi-task) |
|---|---|---|---|---|
| Sequence length | 16 | 16 | 4 | 4 |
| Frame skip | 1 | 1 | 4 | 4 |
| Latent size | $8 \times 8$ | $16 \times 16$ | $16 \times 16$ | $16 \times 16$ |
| Number of classes | 16 | 8 | n/a | 34 |
| Batch size | 32 | 32 | 32 | 32 |
| Learning rate | $10^{-4}$ | $10^{-4}$ | $10^{-4}$ | $10^{-4}$ |
| Learning rate schedule | constant | constant | constant | constant |
| Training steps | 500k | 500k | 500k | 500k |
| Adam $(\beta_1, \beta_2)$ | (0.9, 0.999) | (0.9, 0.999) | (0.9, 0.999) | (0.9, 0.999) |
| Hidden dim | 256 | 512 | 256 | 512 |
| Feedforward dim | 1024 | 2048 | 1024 | 2048 |
| Number of heads | 8 | 8 | 8 | 8 |
| Number of layers | 8 | 8 | 8 | 8 |
| Dropout | 0 | 0 | 0 | 0 |

## A.3 Reinforcement Learning Hyperparameters

Table 3: Hyperparameters and training details for DreamerV3

|  | DMC | Atari | RLBench (single task) | RLBench (multi-task) |
|---|---|---|---|---|
| **General** | | | | |
| Replay Capacity (FIFO) | $10^6$ | $10^6$ | $10^6$ | $10^6$ |
| Start learning (prefill) | 0 | 0 | 0 | 0 |
| Batch Size | 16 | 16 | 16 | 16 |
| Batch length | 64 | 64 | 64 | 64 |
| MLP Size | $2 \times 512$ | $4 \times 1024$ | $4 \times 1024$ | $4 \times 1024$ |
| Activation | LayerNorm + swish | | | |
| **World Model** | | | | |
| RSSM Size | 512 | 4096 | 4096 | 4096 |
| Number of Latents | 32 | 32 | 32 | 32 |
| Classes per Latent | 32 | 32 | 32 | 32 |
| KL Balancing | 0.667 | 0.667 | 0.667 | 0.667 |
| **Actor Critic** | | | | |
| Imagination Horizon | 15 | 15 | 15 | 15 |
| Discount | 0.995 | 0.995 | 0.995 | 0.995 |
| Return Lambda | 0.95 | 0.95 | 0.95 | 0.95 |
| Target Update Interval | 50 | 50 | 50 | 50 |
| **All Optimizers** | | | | |
| Gradient Clipping | 100 | 100 | 100 | 100 |
| Learning Rate | $10^{-4}$ | $10^{-4}$ | $10^{-4}$ | $10^{-4}$ |
| Adam epsilon | $10^{-6}$ | $10^{-6}$ | $10^{-6}$ | $10^{-6}$ |
| **Plan2Explore** | | | | |
| Ensemble size | 10 | 10 | 10 | 10 |

Table 4: Hyperparameters and training details for DrQ

| | DMC |
|---|:---:|
| **Actor Critic** | |
| MLP Size | $256 \times 256$ |
| CNN Features | $32 \times 64 \times 128 \times 256$ |
| CNN Filters | $3 \times 3 \times 3 \times 3$ |
| CNN Strides | $2 \times 2 \times 2 \times 2$ |
| Learning Rate | $3e - 4$ |
| Discount | 0.99 |
| Number of Critics | 2 |
| $\tau$ | 0.005 |
| Init Temperature | 0.1 |
| Augmentations | Random crop |
| **Random Network Distillation** | |
| RND CNN Features | $32 \times 64 \times 64$ |
| RND MLP Size | $512 \times 512$ |
| RND learning rate | $3e - 4$ |
| RND Exploration Weight | 1 |

## A.4 Environment Training Curves

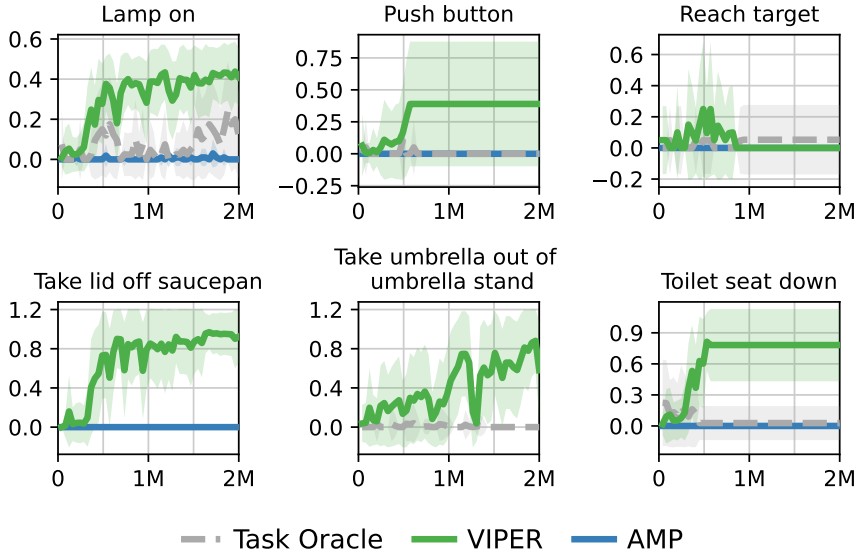

Figure 12: Results across 6 Reinforcement Learning Benchmark tasks, with mean and standard deviation computed across 3 seeds.

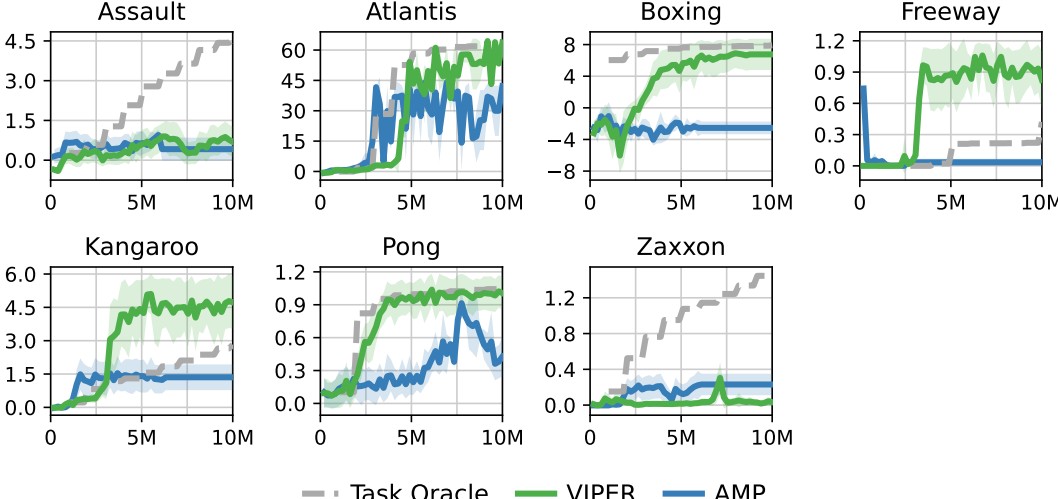

Figure 13: Results across 7 Atari tasks. Scores computed using Human-Normalized Mean across 3 seeds.

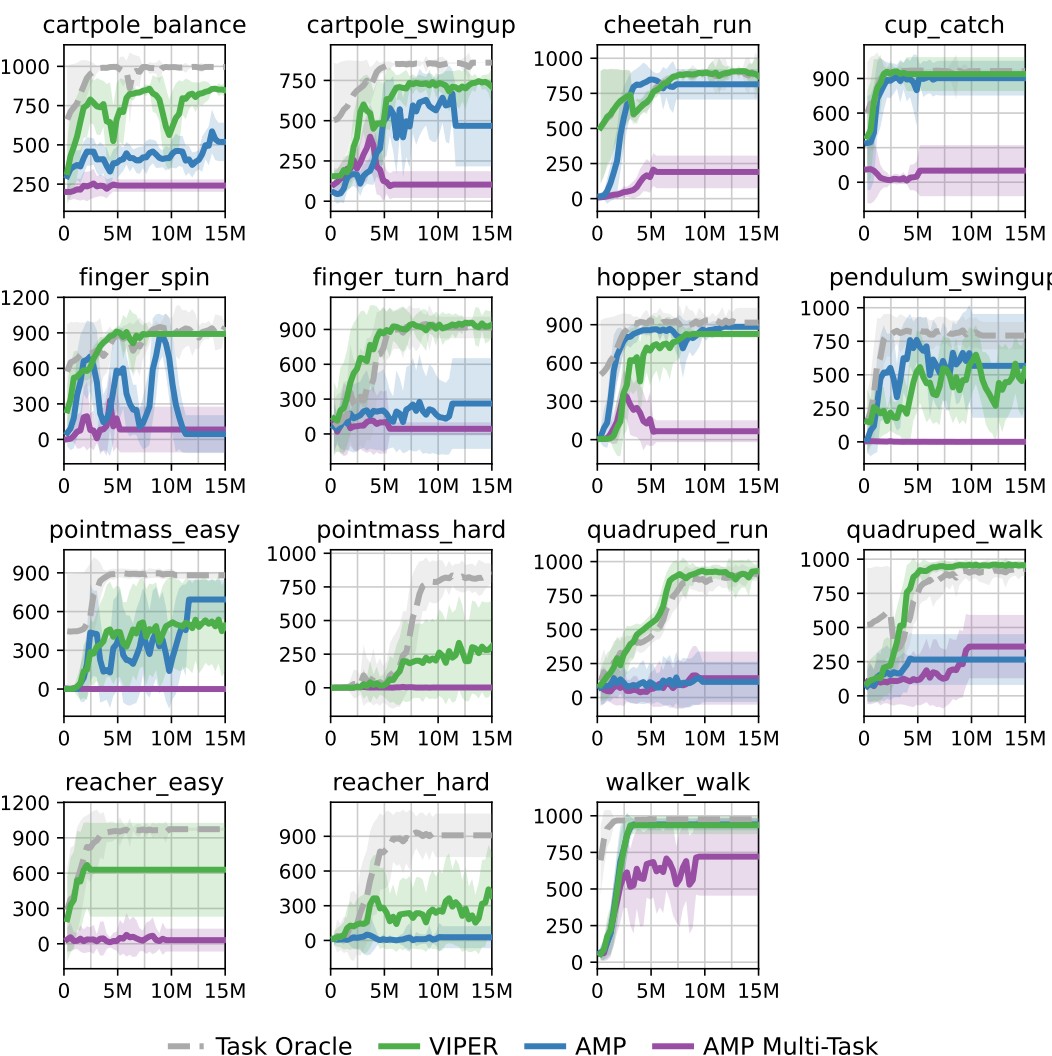

Figure 14: Results across 15 DeepMind Control Suite tasks, with mean and standard deviation computed across 3 seeds. AMP runs were stopped early due to poor performance.

## A.5 Training Environments

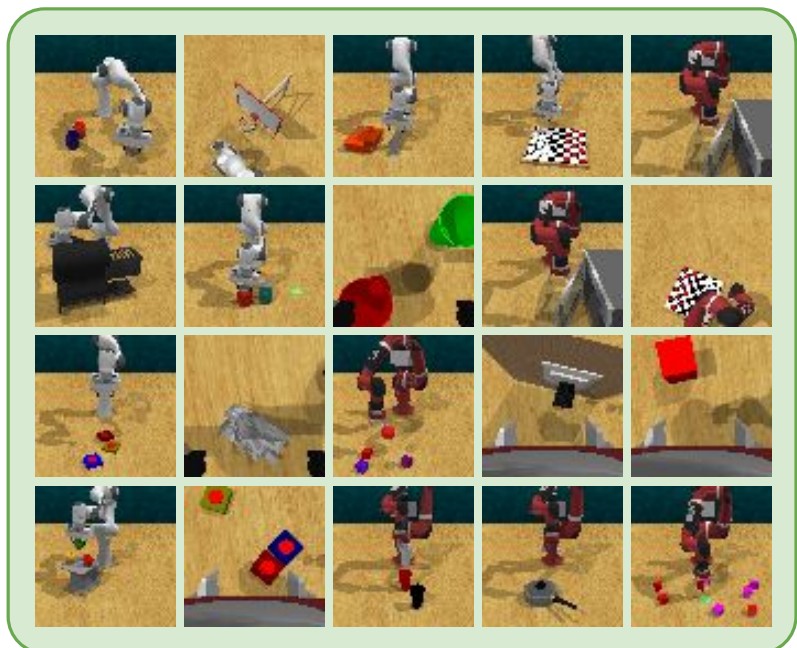

Figure 15: A single autoregressive video model is trained on 30 tasks for the Franka Panda and 23 tasks for the ReThink Robotics Sawyer, using $16 \times 16$ VQCodes and a context length of 4, with frame skip 4.

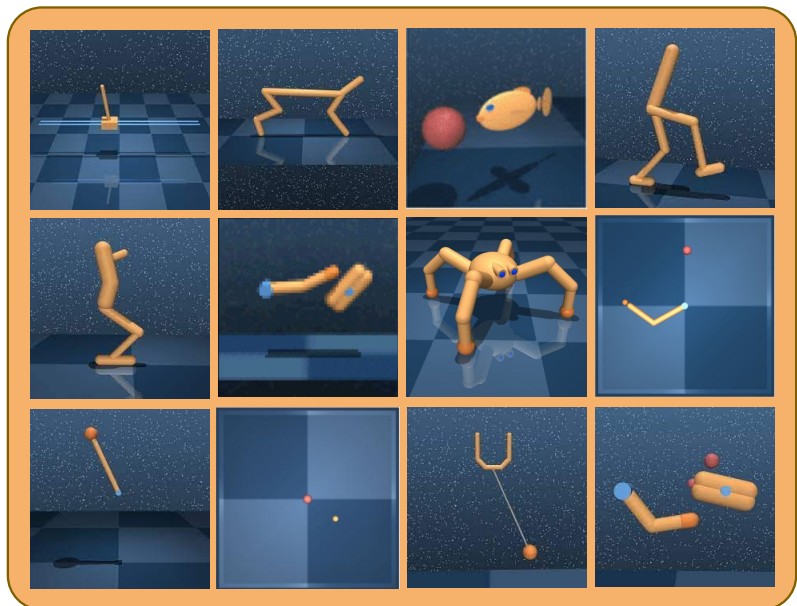

Figure 16: A single task-conditioned autoregressive video model is trained on 15 DeepMind Control Tasks, using $8 \times 8$ VQCodes and a context length of 16.

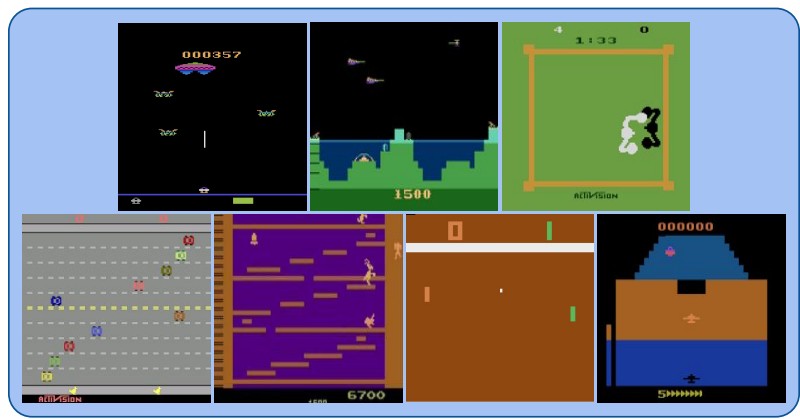

Figure 17: A single autoregressive video model is trained on 7 Atari tasks, using $16 \times 16$ VQCodes and a context length of 16.

## A.6 Visualizing Video Prediction Uncertainty

Video model uncertainty can be visualized by upsampling the conditional likelihoods over VQCodes. Since VQCodes tend to model local features, this provides a useful tool for analyzing which regions of the image the video model is uncertain about. We visualize video prediction uncertainty for three environments below:

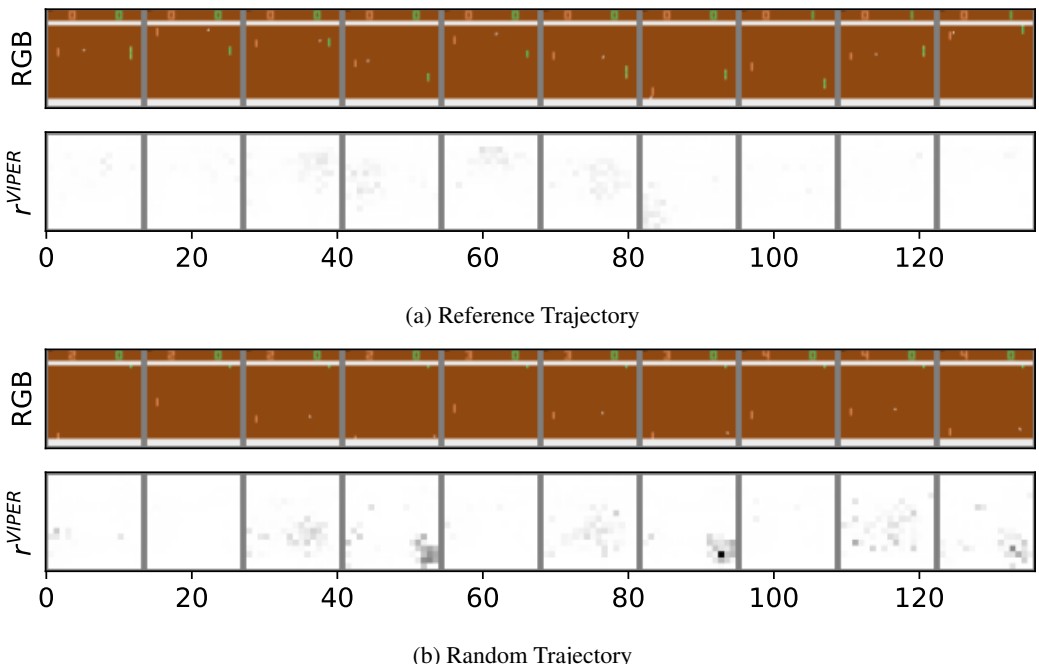

(a) Reference Trajectory

(b) Random Trajectory

Figure 18: Uncertainty visualized for a reference trajectory (top) and a random trajectory (bottom). Brighter values correspond to higher likelihoods. For the random trajectory, the video model assigns lower likelihoods to regions of the image containing the ball. This is especially true for random trajectories, where the video model, trained solely on expert trajectories, cannot accurately predict what happens when the agent plays sub-optimally.

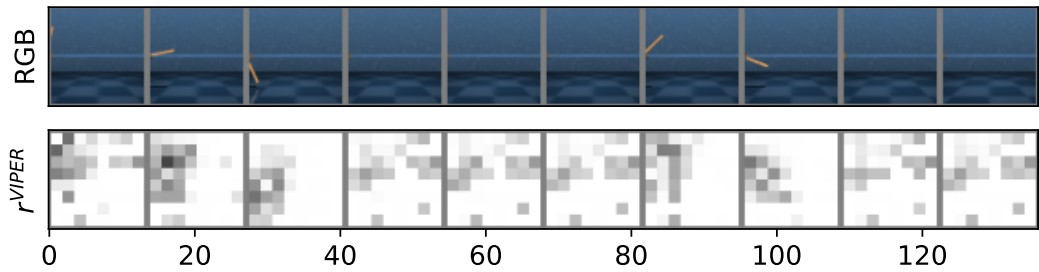

Figure 19: Uncertainty visualized for a random trajectory from the dmc cartpole balance task. Brighter values correspond to higher likelihoods.

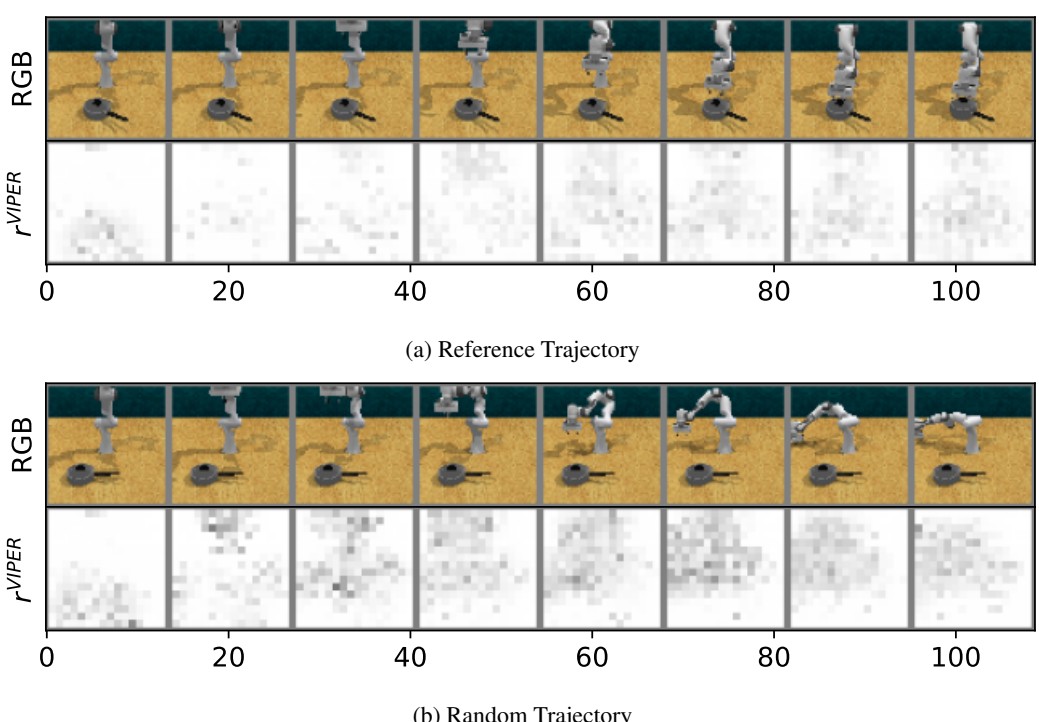

(a) Reference Trajectory

(b) Random Trajectory

Figure 20: Uncertainty visualized for a reference trajectory (top) and a random trajectory (bottom). Brighter values correspond to higher likelihoods. Notice the high uncertainty over the position of objects on the table for the first frame. This corresponds to the unconditional likelihood (with no context). Since the position of the saucepan is randomized for every episode, the video model assigns some uncertainty to this initial configuration. For the random trajectory, there is high uncertainty assigned to the position of the arm since the video model has not seen trajectories that display poor behavior.

## A.7 VIPER Out-of-distribution Analysis

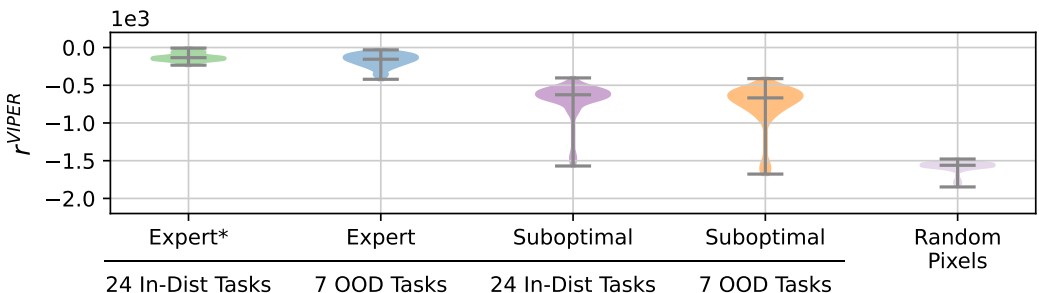

Figure 21: Predicted $r^{\text{VIPER}}$ on various RLBench tasks. We evalute 10 trajectories for each task. *Training data for the VIPER model.

