# OpenReview forum: "Video Prediction Models as Rewards for Reinforcement Learning"
_NeurIPS.cc/2023/Conference — NeurIPS 2023 poster_

### Official Review · Reviewer_1FB4 · 2023-06-27

**Soundness:** 3 good
**Presentation:** 3 good
**Contribution:** 2 fair
**Rating:** 6
**Confidence:** 4

**Summary:**

This paper proposes Video Prediction Rewards (VIPER), a general architecture that extracts reward functions from action-free expert demonstrations. To match the agent's trajectory distribution with the source distribution, VIPER optimizes the agent to maximize a log-likelihood estimated by an auto-regressive video model and an entropy term to encourage exploration.

Experiments on DMC, Atari, and RLBench demonstrate the soundness and efficiency of the reward function extracted by VIPER.

**Strengths:**

VIPER addresses the practical challenge of how to extract reward functions from action-free expert demonstrations in order to optimize our agents, which is useful in settings like self-driving.
VIPER has the following strengths:
- VIPER can extract effective reward functions and thus promote policy optimization in a range of visual control tasks.
- Experiments show that reward functions learned by VIPER can generalize to a variety of tasks and even OOD tasks.

**Weaknesses:**

Although VIPER shows good experiments results, some weaknesses still exist:

- The data efficiency of VIPER seems to be low as it requires nearly 10M data to converge in DMC. Also, it seems that VIPER can not leverage sub-optimal demonstrations, which could be important for improving data efficiency.
- It could be difficult and expensive to acquire a generative video model for real-world tasks, especially with visual distractors.
- Also, I think current tasks are a little bit less challenging, and thus it might be easier to define a reward function than acquire expert demonstrations. Therefore, it could be interesting if we could test VIPER's performance with tasks that are hard to define rewards, e.g. embodied tasks like [Habitat](https://github.com/facebookresearch/habitat-sim) or self-driving platforms.

**Questions:**

- To my understanding, VIPER's setting is similar to Generative Adversarial Imitation Learning (GAIL), while GAIL uses the critic as the surrogate of the distance between the expert trajectory distribution and generated trajectory distribution, VIPER directly estimates the distance (KL divergence) by modeling the log-likelihood with a generative model. I have some reservations regarding the benefits of doing so.

---

> ### Author Rebuttal · Authors · 2023-08-10
>
> We would like to thank the reviewer for their insightful comments. We provide detailed answers to your questions below:
>
> **[Weakness 1-1] The data efficiency of VIPER seems to be low as it requires nearly 10M data to converge in DMC.**
>
> While agents trained with VIPER may require millions of training steps to converge in DMC, agents trained with relevant baselines [A] [B] [C] either take longer to converge, or do not converge at all (additional baselines can be found in Figure A of the PDF in the global rebuttal). Additionally, VIPER outperforms learning from sparse task rewards for all evaluated RLBench tasks (Figure 10, supplementary material) and some Atari tasks (Figure 11, supplementary material). This is quite a promising result, as we can leverage videos to successfully complete tasks which cannot be learned from the sparse task reward. Additionally, VIPER exhibits generalization to a specific cross-embodiment manipulation setting (Figure 7 & 8 of the original submission).
>
> **[Weakness 1-2] Also, it seems that VIPER can not leverage sub-optimal demonstrations, which could be important for improving data efficiency.**
>
> We have conducted additional experiments to verify this (Figure C in the global response PDF). We found that VIPER can still extract reward signal from suboptimal trajectories, which achieve 50-75% return of the expert trajectories. Additionally, we observed faster convergence when training from sparse task rewards in conjunction with rewards extracted from a VIPER model trained on suboptimal data.
>
> **[Weakness 2] It could be difficult and expensive to acquire a generative video model for real-world tasks, especially with visual distractors.**
>
> Figure D of the PDF in the global rebuttal shows further ablations of running VIPER trained on progressively smaller fractions of data, where performance starts degrading at around 10% of the original training data, which amounts to 10 videos in DMC and 10 videos in RLBench. The policy learns less efficient but still meaningful behavior even when VIPER is trained on only 1 expert video.
>
> Thank you for pointing this out. We believe that a promising future direction for VIPER is to leverage pretrained large-scale video generation models, as mentioned in the conclusion section. Doing so relaxes the data requirements for training from scratch, and allows the video model to better generalize to potentially unseen tasks and visual distractors. UniPi [D] shows that this is possible by fine-tuning a large video model on a set of task-specific robotics data, where the resulting model can generate accurate video predictions starting from novel task settings. Therefore, in the VIPER setting, such a model can potentially be used as a generalizable reward function for VIPER training on a diverse range of tasks.
>
> **[Weakness 3] Also, I think current tasks are a little bit less challenging, and thus it might be easier to define a reward function than acquire expert demonstrations. Therefore, it could be interesting if we could test VIPER's performance with tasks that are hard to define rewards, e.g. embodied tasks like Habitat or self-driving platforms.**
>
> Thank you for suggesting interesting and challenging environments to test VIPER. As much as we are excited about the application of VIPER to more challenging environments, we would like to highlight that it is not trivial to define rewards for the robotic manipulation tasks in RLBench. As can be seen in Figure 3 (right) and Figure 10, DreamerV3 failed to learn all 6 tasks with the oracle task rewards whereas VIPER were able to provide more informative rewards. Additionally, The authors consider DMC and Atari to be an important set of benchmarks to consider for this work, as they are a popular choice for evaluating RL algorithms, and make it easy to compare methods. We will leave examining VIPER on more challenging domains, such as Habitat and self-driving, to future work.
>
> **[Q1] To my understanding, VIPER's setting is similar to Generative Adversarial Imitation Learning (GAIL)... I have some reservations regarding the benefits of doing so.**
>
> As the reviewer mentioned, adversarial imitation learning approaches (GAIL, GAIfO, and AMP) have a similar setting with VIPER. However, adversarial training is notoriously prone to mode collapse, exhibits poor generalization properties, and is often challenging to stabilize. Please refer to [E, F] for more thorough discussion about challenges in adversarial training.
>
> VIPER does not formulate KL Divergence minimization as a two-player game. Instead, VIPER treats the video prediction model as a pre-trained and fixed reward model which the RL agent is trained to maximize. As such, VIPER does not exhibit mode collapse. On the other hand, Figure 3 of the original paper and Figure A in the global rebuttal show that VIPER outperforms the adversarial imitation learning baselines (AMP and GAIfO) both in the single and multi-task setting. For example, the VIPER model trained on 16 DMC tasks provides a good reward signal for downstream policy learning, while the corresponding adversarial imitation learning baselines trained on 16 DMC tasks exhibit mode collapse and do not enable policy learning.
>
> ---
>
> We would like to thank the reviewers for their thoughtful comments. Please let us know if there are any concerns preventing you from raising your score.
>
> [A] AMP: Adversarial Motion Priors for Stylized Physics-Based Character Control. Peng et al. SIGGRAPH 2021
>
> [B] Generative adversarial imitation from observation. Torabi et al. arXiv 2018
>
> [C] Behavioral Cloning from Observation. Torabi et al. IJCAI 2018
>
> [D] Learning Universal Policies via Text-Guided Video Generation. Du et al. arXiv 2023
>
> [E] Learning Robust Rewards with Adversarial Inverse Reinforcement Learning. Fu et al. ICLR 2018
>
> [F] Large Scale GAN Training for High Fidelity Natural Image Synthesis. Brock et al. ICLR 2019

---

> > ### Comment · Reviewer_1FB4 · 2023-08-20
> >
> > Thanks for your reply, the additional experiments on sub-optimal trajectories and comparisons with GAIL baselines partially address my concerns. However, I'm still not totally convinced of the advantage of estimating the distance (KL divergence) by modeling the log-likelihood with a generative model, so I'll keep my score.

---

> > > ### Author Response · Authors · 2023-08-21
> > > **Reply to Reviewer 1FB4**
> > >
> > > We would like to thank the reviewer for engaging in the author-reviewer discussion period. We would like to clarify and re-iterate VIPER’s strengths.
> > > - VIPER uses a **powerful video prediction architecture**, which enables **stable training** from **diverse multi-task data** and modeling the log likelihood of the expert video distribution.
> > > - Our reward formulation captures distributions over **trajectories** instead of transitions, which leads to more informative rewards.
> > >
> > > Throughout our paper and rebuttal, we have provided evidence to support these advantages.
> > >
> > > - VIPER outperforms the baselines (AMP, GAIfO, BCO) in the single task setting. More importantly, VIPER drastically outperforms them in the multi-task setting (Figure 3 in paper, Figure A in global response). This indicates that VIPER does not suffer from mode collapse (as do adversarial methods) and exhibits better training stability.
> > > - VIPER exhibits a specific form of generalization, whereby the learned reward model can generalize to unseen embodiments for certain tasks (Figure 8 in paper, Figure B in global response).
> > > - VIPER does not require very large expert video data to learn a performant reward model. In fact, VIPER can learn a meaningful reward model from as little as **1 trajectory** (Figure D).
> > >
> > > We again thank the reviewer for their comments

---

### Official Review · Reviewer_qQw6 · 2023-06-30

**Soundness:** 3 good
**Presentation:** 3 good
**Contribution:** 3 good
**Rating:** 6
**Confidence:** 3

**Summary:**

The authors present Video Prediction Rewards, an algorithm that leverages transformer-based video prediction models as action-free reward signals for reinforcement learning. The reward induced by this video prediction model incentivizes the agent to find the most likely trajectory under the expert video distribution. By further incorporating some exploration rewards, such as RND, the proposed method obtains good performance across a wide range of DMC, Atari, and RLBench tasks.

**Strengths:**

The paper is well written and easy to read.
The authors aim to address a crucial problem in reinforcement learning, i.e., the reward function design. The authors propose a concise method, and experimental results also validate the effectiveness of the approach.


**Weaknesses:**

I'm concerned about the problem of out-of-distribution. Can the pre-trained video prediction models accurately evaluate unseen behaviours?

**Questions:**

See the weakness

---

> ### Author Rebuttal · Authors · 2023-08-10
>
> We are happy that the reviewer recognized the importance of learning reward functions for RL and found our method concise and effective across a wide range of tasks. The PDF in the global rebuttal contains:
>
> - Additional experiments with more baselines [A, B] (Figure A)
> - Further analysis of OOD generalization (Figure B), and an analysis of VIPER models trained on sub-optimal data (Figure C)
> - An analysis of VIPER performance as a function of dataset size (Figure D).
>
> ---
>
> Please let us know if there are any concerns preventing you from raising your score. If there are any suggestions or concerns that we could address to strengthen the paper further, we would be glad to engage during the discussion period.
>
> [A] Generative adversarial imitation from observation. Torabi et al. arXiv 2018
>
> [B] Behavioral Cloning from Observation. Torabi et al. IJCAI 2018

---

> > ### Comment · Reviewer_qQw6 · 2023-08-11
> >
> > I have read the author's response, and I maintain my original score.

---

> > > ### Author Response · Authors · 2023-08-13
> > > **Reply to reviewer qQw6**
> > >
> > > We are curious to hear what improvements the reviewer would like to see which will improve the paper.
> > >
> > > **[Re: Weakness 1 (after revision from reviewer)] I'm concerned about the problem of out-of-distribution. Can the pre-trained video prediction models accurately evaluate unseen behaviours?”**
> > >
> > > Thank you for pointing this out. Experiments conducted during the Author response period extended the scope of our original analysis to include more in-distribution and out-of-distribution environments, and more behaviors which were not seen by the VIPER model during training. We found that VIPER consistently predicts higher rewards for expert behaviors (which successfully complete the task) than for suboptimal behaviors (e.g. random behaviors, behaviors which don’t successfully complete the task), regardless of the environment being in-distribution or out-of-distribution (Figure B in the global response PDF).
> > >
> > > Additionally, Figure D in the global response PDF indicates that VIPER models trained on **1 trajectory** can provide rewards that lead to less efficient, yet meaningful behaviors. We believe this requires generalization of the learned VIPER model, since only one behavior was seen during video model training yet the reward function is evaluated on many behaviors during RL training.
> > >
> > > Finally, previous work in video generation [A, B, C] and robotics [D] has studied the ability of video generation models to generalize beyond the training distribution, yielding promising results. We find that VIPER benefits from video prediction generalization, and learns a reward function which generalizes favorably to unseen environments, enabling learning of performant RL policies (Figure 7, 8 in the original paper).
> > >
> > > ---
> > >
> > > [A] Imagen Video: High Definition Video Generation with Diffusion Models. Ho et al. arXiv 2022
> > >
> > > [B] Phenaki: Variable Length Video Generation From Open Domain Textual Description. Villegas et al. arXiv 2022
> > >
> > > [C] Make-A-Video: Text-to-Video Generation without Text-Video Data. Singer et al. arXiv 2022
> > >
> > > [D] Learning Universal Policies via Text-Guided Video Generation. Du et al. arXiv 2023

---

### Official Review · Reviewer_d6Ah · 2023-06-30

**Soundness:** 3 good
**Presentation:** 3 good
**Contribution:** 3 good
**Rating:** 7
**Confidence:** 4

**Summary:**

This paper proposes a learning-from-observation algorithm that builds a reward based on a video predictor trained from action-free expert videos. Experimental results show that online reinforcement learning algorithms can learn a working policy from their reward only effectively. This paper contains rich and informative ablation studies and analyses to verify their design choices.

**Strengths:**

1.	The paper writing is clear and easy to understand
2.	Distilling knowledge from action-free videos to policies is a promising future direction for robotics.
3.	Experiments are rich. The authors test their method with two different online RL methods, two different exploration losses, three task domains, and three different video prediction models.

**Weaknesses:**

The experiments on the generalization ability (Sec. 4.3) are not convincing enough. The video prediction model in Sec.4.3 is trained with 23 Rethink-robot-arm tasks and 30 Franka-robot-arm tasks. There should be dozens of OOD arm/task combinations that can be evaluated. However, according to L300, we only see the performance on only ONE OOD combination. How is the performance on other OOD combinations? Besides, the learning curve in Fig.8 doesn’t include a task oracle like other experiments in the paper. So we also don’t know how good the OOD performance is. Therefore, I think the third contribution of this paper, “VIPER generalizes to different environments”, is not well-supported.

**Questions:**

1.	How good is the generalization ability of VIPER? We definitely need evaluations on more OOD combinations to support the statement in the third contribution.
2.	For Fig.8, which OOD combination the curve shows? In addition, this curve doesn’t include an error bar like other experiments in the paper.

**Limitations:**

The authors listed and discussed the limitations including the lack of in-domain expert data in the real world, the sub-optimal performance with stochastic data, and the sensitive performance to the VQCode size and context length.

---

> ### Author Rebuttal · Authors · 2023-08-10
>
> We thank reviewer d6Ah for their thoughtful comments, and provide detailed answers to their questions below.
>
> **[Q1] How good is the generalization ability of VIPER? We definitely need evaluations on more OOD combinations to support the statement in the third contribution.**
>
> Thank you for pointing this out. We agree that a more thorough understanding of out-of-distribution robustness is helpful to better understand the performance of VIPER on OOD data. Thus, we have additionally evaluated VIPER on both in-distribution (24 training tasks) and out-of-distribution (7 held-out tasks) data ranging from suboptimal to expert (Figure B of the PDF in the global rebuttal).
>
> Our analysis suggests that VIPER learns a reward model that is not adversely affected by OOD training samples, and video prediction rewards are consistently higher for expert trajectories, even for tasks which were not seen during video model training. On the other hand, adversarial imitation learning baselines, AMP [A] and GAIfO [B], exhibit poor generalization properties [C] due to its instability in training and mode collapse, as explained in [D - Section 4.2]. In summary, VIPER exhibits generalization to OOD states and tasks which were not encountered during video model training. Given the relatively small size of the training dataset, we were excited to see the video model generalize to novel arm/task combinations. Scaling up the video training dataset and model size may yield stronger generalization results.
>
> We are also in the process of running RL experiments for more OOD arm/task combinations. We will include these results in the revised version.
>
>
> **[Q2] For Fig.8, which OOD combination the curve shows? In addition, this curve doesn’t include an error bar like other experiments in the paper.**
>
> The left plot in Figure 8 shows a training run for the OOD arm/task combination visualized in Figure 7: the panda performing take_lid_off_saucepan. We will update the main text of the paper to better reflect this.We only included one trajectory for this visualization / training curve, but will update the revised text to include more RL training runs. Figure B the revised PDF demonstrates that VIPER provides meaningful rewards on 7 OOD tasks on the panda: push_button, push_button(s), toilet_seat_down, reach_target, take_lid_off_saucepan, take_umbrella_out_of_stand, reach_target.
>
> ---
>
> We would like to thank the reviewer once again for their insightful comments, and look forward to engaging with them in the reviewer-author discussion period.
>
> [A] AMP: Adversarial Motion Priors for Stylized Physics-Based Character Control. Peng et al. SIGGRAPH 2021
>
> [B] Generative adversarial imitation from observation. Torabi et al. arXiv 2018
>
> [C] Learning Robust Rewards with Adversarial Inverse Reinforcement Learning. Fu et al. arXiv 2018
>
> [D] Large Scale GAN Training for High Fidelity Natural Image Synthesis. Brock et al. arXiv 2019

---

> > ### Comment · Reviewer_d6Ah · 2023-08-18
> >
> > Thanks for the response to my questions. I think the new OOD experiments are good support for the effectiveness of this method. Besides, I appreciate the new imitation learning baseline results. I think this paper is a good exploration of a promising direction to learn a model that provides rewards and feedback. I'll increase my score.

---

> > > ### Author Response · Authors · 2023-08-18
> > > **Reply to reviewer d6Ah**
> > >
> > > We kindly thank reviewer d6Ah for increasing their score; however, the score hasn't changed yet. Hope to see the increased score in the review soon!

---

### Official Review · Reviewer_SQjv · 2023-07-12

**Soundness:** 3 good
**Presentation:** 2 fair
**Contribution:** 2 fair
**Rating:** 5
**Confidence:** 5

**Summary:**

This paper proposes to use prediction likelihoods from autoregressive video models as reward functions to train reinforcement learning agents. Specifically, the conditional likelihood $\log p(x_{t+1}|x_{1:t})$ is augmented with an exploration reward to avoid suboptimal behavior. The authors conduct extensive experiments and show that likelihoods of autoregressive video models can be effective for reward specification. They also show that in certain cases the video models can generalize to unseen task domains, encouraging satisfying behaviors. Their ablation study compares different video models, different exploration objectives, and different context lengths.

**Strengths:**

Originality: Though the idea of using likelihood of states/observations as a reward is not novel, taking the temporal coherence into consideration with an autoregressive factorization is novel at my end.

Quality: This work is strong in its efforts in extensive experiments.

Clarity: This paper is straightforward to follow. The narrative is very intuitive. Experimental details are very well documented.

Significance: Learning memory-based reward function with sequence modeling is an interesting direction to explore given the current advances of generative models.

**Weaknesses:**

In spite of the impressive amount of experiments presented in this work, one fundamental problem unresolved in this work is why the autoregressive likelihood, which is inherently non-Markov, can work with general RL algorithms, in which TD learning strongly depends on Markovian rewards. Latent-space model-based RL methods such as Dreamer used in this work are too particular because the latent-space modeling may resolve the limitation of TD learning as a byproduct. This means the empirical result from this work cannot be trivially generalized to other RL methods, rendering the thesis statement an overclaim.

**Questions:**

Apart from the question I raised in Weakness, I hope the authors would also like to resolve my concerns in Section 4.3.

While the paper claimed that specifying reward functions with video models can generalize to OOD tasks, Section 4.3 only demonstrate a particular case where there is a recombination of robot and objects to be manipulated. Is it possible to make the evaluation of generalization more systematic? I guess readers may be more interested in a discussion of what "types" of generalization are possible to eliminate the influence of particularity.

**Limitations:**

As stated in Weakness, there is a technical limitation of the proposed method that the authors do not seem to notice. Other limitations are well documented in Section 5.

---

> ### Author Rebuttal · Authors · 2023-08-10
>
> We would like to thank the reviewer for their thoughtful comments. We provide detailed answers to your questions below:
>
> **[Weaknesses]  one fundamental problem unresolved in this work is why the autoregressive likelihood, which is inherently non-Markov, can work with general RL algorithms, in which TD learning strongly depends on Markovian rewards.**
>
> Thank you for pointing out the non-Markovian nature of VIPER. We agree that non-Markovian VIPER rewards could be an issue in TD learning. However, in practice, state-of-the-art RL algorithms based on TD learning can handle many non-Markovian rewards (in the simplest case by leveraging frame stacking), such as the rewards provided by adversarial imitation learning [A, B], Minecraft [C], Crafter [D], and many more POMDP environments. We will make this point clear in the revised paper. Despite the non-Markovian property of  VIPER, we show that VIPER works well with a model-free off-policy RL algorithm, DrQ, which does not use a recurrent latent state model, in Figure 3 of the original paper.
>
> **[Q1]: Is it possible to make the evaluation of generalization more systematic? I guess readers may be more interested in a discussion of what "types" of generalization are possible to eliminate the influence of particularity.**
>
> In this work, we focus mainly on the application of video prediction models to reward specification and show one specific type of generalization: cross-embodiment generalization, whereby the video prediction model is trained on videos of tasks for the sawyer and panda arm, with some tasks held out for the panda arm, requiring that the model generalize and provide accurate rollouts for the held out tasks.  We have extended this analysis to include more tasks (Figure B of the PDF in the global rebuttal). Our analysis suggests that VIPER learns a reward model that is not adversely affected by OOD training samples, and exhibits generalization to OOD tasks which were not encountered during video model training. Video prediction rewards are consistently higher for expert trajectories, even for tasks which were not seen during video model training.
>
> Related works such as UniPi [E] conduct more extensive combinatorial generalization experiments that evidence the current capabilities of large video prediction models. We believe scaling up the training dataset and video model size should lead to a more general reward model.
>
> ---
>
> We thank the reviewer again for their insightful comments, and look forward to engaging with them in the reviewer-author discussion period. Please let us know if there are any concerns preventing you from raising your score.
>
>
> [A] AMP: Adversarial Motion Priors for Stylized Physics-Based Character Control. Peng et al. SIGGRAPH 2021
>
> [B] Generative adversarial imitation from observation. Torabi et al. arXiv 2018
>
> [C] MineRL: A Large-Scale Dataset of Minecraft Demonstrations. Guss et al. IJCAI 2019
>
> [D] Benchmarking the Spectrum of Agent Capabilities, Hafner. ICLR 2022
>
> [E] Learning Universal Policies via Text-Guided Video Generation. Du et al. arXiv 2023

---

> > ### Comment · Reviewer_SQjv · 2023-08-18
> >
> > Thank the authors for their response and the additional experiments.
> >
> > Regarding the issue of non-Markovianness in the RL problem, I recommend that the authors consider including a more in-depth discussion about the potential risks associated with disregarding this aspect in their revision. While the authors have highlighted specific instances of empirical success, there is a need to ascertain whether these accomplishments pertain to tasks with minimal "aliasing," where the underlying limitations might not be fully exposed.
> >
> > Regarding the notably ambiguous term "OOD generalization," I think a more cautious narrative is advisable. Perhaps opting for more precise terminology, such as "cross-embodiment generalization," could enhance clarity. Ultimately, the monumental challenge of OOD generalization remains unresolved within the scope of this work, doesn't it?

---

> > > ### Author Response · Authors · 2023-08-18
> > > **Reply to reviewer SQjv**
> > >
> > > **[Non-markovian rewards & aliasing]**
> > >
> > > The authors thank the reviewer for their comments, and agree with their suggestions. The revised manuscript will acknowledge that the reward is non-markovian, discuss potential risks, and consider different approaches to handling the non-markovian reward provided by VIPER.
> > >  We agree that evaluating alternative environments to probe the issue of aliasing is an interesting analysis to perform. But, we will leave this analysis to future work.
> > >
> > > **[Generalization]**
> > >
> > > Yes, tackling the fundamental problem of OOD generalization is out of the scope of this paper, and we agree that the paper’s generalization discussion warrants precise terminology. In the original paper we cautiously refer to our experiments as evaluating “cross-embodiment generalization” in the abstract, contributions, experiments sections. The revised manuscript will contain additional clarification, as well as results from the experiments found in the global response.
> > >
> > > We would like to thank Reviewer SQjv again for engaging in discussion and providing constructive feedback. Please let us know if you have additional questions or suggestions.

---

### Official Review · Reviewer_BsvD · 2023-07-24

**Soundness:** 2 fair
**Presentation:** 3 good
**Contribution:** 2 fair
**Rating:** 5
**Confidence:** 3

**Summary:**

The paper proposes to use a large video-prediction model for learning a reward model for RL. The agent's performance is evaluated on a total of 8 envs from 3 benchmarks: DMC, Atari, and RLBench. The paper argues that the proposed model also generalizes to different environments for which no training data was provided, enabling cross-embodiment generalization for tabletop manipulation.

**Strengths:**

* The paper is well-written and easy to follow.
* Section 4 is very well organized. It starts by asking base questions like "can VIPER provide an adequate learning signal for solving a variety of tasks?" before jumping on to the evaluation of the performance of the RL algorithm.

**Weaknesses:**

Limited baselines: The paper compares with just 1 baseline (the second "baseline" is more of an ablation of the first baseline). e.g. there is https://sites.google.com/view/vip-rl that claims to provide "dense visual reward". The paper itself lists a bunch of baselines (in the related work) but does not compare to them.

Limited ablations: See questions.

Overall, I think the paper is interesting but I want to see performance improvement over a bunch of baselines and some ablations. I would encourage the authors to engage during the rebuttal period.

**Questions:**

1. In line 128, the paper states "For example, when flipping a weighted coin with p(heads = 0.6) 1000 times, typical sequences will count roughly 600 heads and 400 tails, in contrast to the most probable sequence of 1000 heads that will basically never be seen in practice". Could the authors explain why is the sequence of 1000 heads the most probable one ?
2. Does the algorithm work with good trajectories as well or does it need access to expert trajectories? e.g. in line 172, what if they were using the top 650 to top 550 episodes, in place of the top 100 episodes. This would make for an useful ablation experiment.
3. Arent the video datasets "too small"? Given that the video models are trained for hundreds of thousands of updates, I wonder if the video models are drastically overfitting, leading to (i) the learned policies not showing any diverse behaviours and (ii) the learned policies failing with stochastic envs. This would make for another useful ablation experiment.
4. Line 201 states that TPUs were used for training while 226 states that GPUs were used for training. Which is it :)

---

> ### Author Rebuttal · Authors · 2023-08-10
>
> We would like to thank the reviewer for their helpful comments. We provide detailed answers to their questions below.
>
> **[Weaknesses] Overall, I think the paper is interesting but I want to see performance improvement over a bunch of baselines and some ablations.**
>
> We have included additional baselines, GAIfO [B] and BCO [C] in Figure A of the PDF in the global rebuttal. The results show that BCO fails to learn from pixels in all three task suites and GAIfO performs worse than VIPER but similarly to the AMP baseline [A].
>
> - GAIfO is similar to the AMP baseline in the original submission but optimizes KL Divergence instead of Pearson Divergence. Both perform similarly in our experiments and are outperformed by VIPER. Both GAIfO and AMP are prone to mode collapse in the multi-task setting, whereas VIPER scales favorably with the number of tasks.
>
> - We had originally compared VIPER to BCO but found that it performed poorly from pixels even with extensive tuning, and thus we did not include it originally (Figure A of global response PDF). Further investigation indicates that its inverse dynamics model struggles with the hard generalization challenge of labeling the expert videos with actions after being only trained on the suboptimal trajectories encountered by the agent so far, especially when trained from pixels.
>
> **[Q1] Why is the sequence of 1000 heads the most probable one?**
>
> We clarified this thought experiment in the paper as discussed here. Flipping a biased coin with $p(\text{heads}=0.6)$ 1000 times usually yields sequences with roughly 600 heads and 400 tails. Those sequences roughly have probability $0.6^{600} \times 0.4^{400} = 5e^{-293}$. In contrast, the sequence of 1000 heads has the probability of $0.6^{1000} = 1e^{-222}$, which is  more likely (and "degenerate") than the sequence with 600 heads, $5e^{-293}$.
>
> Similarly, if the policy aims to replicate the most probable trajectory under the video model, its behavior may be very different from the reference videos the video model was trained on and possibly result in degenerate behavior. We will make this clearer in the revised version.
>
> **[Q2] Does the algorithm work with good trajectories as well or does it need access to expert trajectories?**
>
> Given that the RL objective is to maximize the likelihood of agent trajectories under the video model, when trained with good trajectories (we interpreted it as suboptimal trajectories), the policy would learn behaviors similar to those in the distribution of good trajectories.
>
> We have conducted additional experiments to verify this (Figure C of the PDF in the global rebuttal). We found that VIPER can extract reward signals from good, but suboptimal, trajectories, which achieve 50-75% the return of the expert trajectories (this varies between tasks, but these trajectories achieve between 500 and 800 undiscounted return). Additionally, we observed faster convergence when training from sparse task rewards in conjunction with rewards extracted from a VIPER model trained on “good” data.
>
> **[Q3] Aren’t the video datasets “too small”? … I wonder if the video models are drastically overfitting… leading to (i) the learned policies not showing any diverse behaviours and (ii) the learned policies failing with stochastic envs**
>
> We did not observe much overfitting as our video models were generally small (~10M parameters). Depending on the environment, training curves for test set loss generally converged to a point at or slightly higher than the train set loss. We point the reviewer to Figures 7 & 8 in the original submission, where we probe the video model’s ability to generalize to an out-of-distribution task. Figure 8 shows that this generalization enables VIPER to provide a good reward signal for policy learning in an OOD environment.
>
> To better understand performance of VIPER on OOD data, we have additionally evaluated VIPER rewards on both in-distribution and out-of-distribution data ranging from suboptimal to expert (Figure B of the PDF in the global rebuttal). Our analysis suggests that VIPER learns a reward model that is not adversely affected by OOD training samples, and video prediction rewards are consistently higher for expert trajectories, even for tasks which were not seen during video model training. In summary, VIPER exhibits generalization to OOD states and tasks which were not encountered during video model training.
>
> Regarding diversity, we note that RLBench demonstrations cover multiple modes because they are gathered by a custom random shooting trajectory optimizer. Additionally, each RLBench environment is randomized for each demo. Additionally, even though the DMC and Atari environments are deterministic, the policies used to gather the demonstrations are stochastic, leading to diverse behaviors.
>
> **[Q4] Were TPUs or GPUs used?**
>
> Sorry for the confusion. All video models were trained using TPUs (v3), and all RL models were trained on GPUs (V100, A100). In general, both video models and RL models can be trained on either TPUs or GPUs. We will clarify this in the revised version.
>
> ---
>
> We would be delighted if you could confirm which of your concerns we resolved and whether there are any issues remaining that might prevent acceptance of the paper.
>
>
>
>
> [A] AMP: Adversarial Motion Priors for Stylized Physics-Based Character Control. Peng et al. SIGGRAPH 2021
>
> [B] Generative adversarial imitation from observation. Torabi et al. arXiv 2018
>
> [C] Behavioral Cloning from Observation. Torabi et al. IJCAI 2018

---

> > ### Comment · Reviewer_BsvD · 2023-08-20
> > **Thank you for the clarifications!**
> >
> > Thank you authors for sharing your responses.
> >
> > > We have included additional baselines, GAIfO [B] and BCO [C]
> >
> > Thank you. GAlfO is quite similar to AMP. I would have preferred to see a baseline that is not so related to AMP though I do understand that time is constrained during the rebuttal period.
> >
> > > Why is the sequence of 1000 heads the most probable one?
> >
> > I understand the point the authors are making (that a sequence with all heads is less likely to occur than a sequence with 60% heads and 40% tails) but it is framed in a confusing way. The authors should clarify that the sequence with 1000 heads is the most probable sequence among all sequence of coin flips (i.e. make it explicit that the authors are considering the ordering of heads and tails in a sequence This is not clear because of the mention of 600 heads and 400 tails). Anyways, this is a minor thing and I am confident that the authors will explain it better in the final draft.
> >
> > I have read the other replies and I am increasing my score to 5.

---

> > > ### Author Response · Authors · 2023-08-20
> > > **Reply to reviewer BsvD**
> > >
> > > We would like to thank the reviewer for their insightful comments, and for increasing their score. Their feedback was invaluable.
> > >
> > > **[I would have preferred to see a baseline that is not so related to AMP]** Thank you for noting this. We also included a Behavioral Cloning from Observation (BCO) baseline [A], which is quite different from AMP and GAIL in that it learns an inverse dynamics model from the policy rollouts which is then used to predict actions for the expert videos (Figure A). We chose AMP, GAIfO, and BCO as baselines because these exemplary works represent interesting solutions to the problem of imitation learning from videos. However, we are working to include more imitation learning baselines into the final version of the paper, including VIP [B].
> > >
> > > **[Re: Typical sequence discussion]** We strongly agree with the reviewer that a clearer explanation of this concept is necessary. As such, we have simplified the explanation in the main text and added further clarification as a footnote so as to not confuse the reader.
> > >
> > > ---
> > >
> > > We again thank the reviewer for engaging in the author-reviewer discussion period. Please let us know if you have any additional concerns or questions.
> > >
> > > [A] Behavioral Cloning from Observation. Torabi et al. IJCAI 2018
> > > [B]  VIP: Towards Universal Visual Reward and Representation via Value-Implicit Pre-Training. Ma et al. ICLR 2023

---

### Official Review · Reviewer_Zre9 · 2023-07-25

**Soundness:** 2 fair
**Presentation:** 4 excellent
**Contribution:** 2 fair
**Rating:** 3
**Confidence:** 4

**Summary:**

This paper proposes a simple method that uses a pre-trained video prediction model to provide rewards for online RL. The design includes using VQ-GAN to encode discrete embeddings and incorporating an exploration bonus (opt for Plan2Explore and RND). In experiments, the authors also show the learned rewards provide a useful learning signal for online RL.

**Strengths:**

1. The paper is well-written and the idea is clear.
2. The authors make comparisons on multiple tasks.

**Weaknesses:**

1. The effectiveness of the proposed method may be limited if the expert data is scarce.
2. In many imitation learning papers almost only one expert trajectory is needed, however, this paper undoubtedly requires a lot of expert data (to train the video prediction model).
3. Although the authors make comparisons on multiple tasks, there are few baselines. There are many papers on imitation learning that do not make experimental comparisons, e.g. [1, 2, 3, 4, 5].

[1] Optimal Transport for Offline Imitation Learning
[2] Demodice: Offline imitation learning with supplementary imperfect demonstrations
[3] Behavioral cloning from observation
[4] Generative adversarial imitation from observation
[5] CLUE: Calibrated Latent Guidance for Offline Reinforcement Learning

**Questions:**

1. Can the author compare the proposed method to more imitation learning papers?
2. Doesn't the method suffer from the problem of OOD issues when there is very little expert data? Even when a dozen or so pieces of expert data exist, it seems that the OOD problem exists, i.e., the pre-trained video prediction model may falsely overestimate the probability of some behaviors that are not expert behaviors.
3. After I thought deeply about it, I always thought that there is an OOD problem with the method, which is consistent with standard offline RL, as the policy network will make an effort to explore and discover behaviors with a high probability/likelihood, however, these behaviors may be falsely overestimated by the video prediction network.
4. In the main paper, I did not see the results that "VIPER can achieve expert-level control without task rewards on 15 DMC tasks, 6 RLBench tasks, and 7 Atari tasks".
5. In addition, the authors only emphasize achieving expert-level performance and do not compare it to a large number of imitation learning baselines. This tends to raise doubts about the performance of the method, since with enough expert data, simple behavioral cloning can also achieve expert-level performance.

**Limitations:**

The authors briefly discuss the limitations of the paper.

---

> ### Author Rebuttal · Authors · 2023-08-10
>
> Thank you for your constructive feedback. We provide detailed answers to the questions below.
>
> **[Weakness 1 & 2] The effectiveness of the proposed method may be limited if the expert data is scarce. … this paper undoubtedly requires a lot of expert data (to train the video prediction model).**
>
> Figure D in the global response PDF shows an additional ablation study on the dataset size. We find that VIPER achieves high performance even with only 10 expert videos (10% of the original dataset size), and learns a less efficient but still meaningful behavior even with only 1 expert video.
>
> The primary motivation for our work is to derive rewards from video models trained on diverse data rather than a few task-specific demonstrations, as general video models will likely become available in the future. In our paper, we train a single video model per suite (e.g. one video model for 50 RLBench tasks).
>
> While prior works, such as GAIfO [A] and AMP [B], struggle with mode collapse when the data is diverse, VIPER scales well as more data is used. Moreover, by learning from diverse videos, VIPER can exhibit generalization to an unseen tabletop manipulation setting (Section 4.3).
>
> **[Weakness 3 & Q1] Can the author compare the proposed method to more imitation learning papers?**
>
> We have included additional baselines, GAIfO and BCO [C] in Figure A of the PDF in the global response. The results show that BCO fails to learn from pixels in all three task suites and GAIfO performs worse than VIPER but similarly to the AMP baseline.
>
> - GAIfO is similar to the AMP baseline in the original submission but optimizes KL Divergence instead of Pearson Divergence. Both perform similarly in our experiments and are outperformed by VIPER. Both GAIfO and AMP are prone to mode collapse in the multi-task setting, whereas VIPER scales favorably with the number of tasks.
>
> - We had originally compared VIPER to BCO but found that it performed poorly from pixels even with extensive tuning, and thus we did not include it originally (Figure A of global response PDF). Further investigation indicates that its inverse dynamics model struggles with the hard generalization challenge of labeling the expert videos with actions after being only trained on the suboptimal trajectories encountered by the agent so far.
>
> As suggested by Reviewer Zre9, we will include the BCO and GAIfO results in the revised paper.
>
> **[Q2 & Q3] Doesn't the method suffer from the problem of OOD issues when there is very little expert data?**
>
> For the ability of VIPER to learn from very little expert data, such as a single expert video trajectory, we refer to Figure D in the global response PDF (as discussed above).
>
> To show the OOD generalization of VIPER more generally, the original submission includes Figure 8 that shows generalization of the video model to unseen combinations of robot arms and tasks, and successful policy learning based on this reward.
>
> To further address this point, we conducted an additional analysis of the rewards assigned by VIPER to various trajectory distributions after training on only expert videos of in-distribution tasks (Figure B in the global response PDF). Notably, we find that VIPER assigns higher rewards to expert behavior on OOD tasks than to suboptimal and random behaviors on in-distribution tasks. The reward histograms show that VIPER reflects different levels of task performance well, even out of distribution.
>
> **[Q4] In the main paper, I did not see the results that "VIPER can achieve expert-level control without task rewards on 15 DMC tasks, 6 RLBench tasks, and 7 Atari tasks".**
>
> Thank you for pointing this out. As shown in Figure 3 of the original paper, VIPER performs similarly to the agents trained with the ground-truth task rewards on the DMC, RLBench, and Atari tasks **on average**. Specifically, VIPER achieves expert-level performances on 6/6 RLBench tasks (Figure 10), 5/7 Atari tasks (Figure 11), and 8/15 DMC tasks (Figure 12). We will clarify and state this explicitly in the revised paper.
>
> **[Q5] In addition, the authors only emphasize achieving expert-level performance and do not compare it to a large number of imitation learning baselines. This tends to raise doubts about the performance of the method, since with enough expert data, simple behavioral cloning can also achieve expert-level performance.**
>
> We refer to our answer above to **Weakness 3 & Q1**, where we discuss two additional baselines that we added to the paper: GAIfO and BCO (Figure A of the PDF in the global rebuttal). The adversarial methods do not perform as well as VIPER even in the single task setting, and perform poorly in the multi-task setting due to mode collapse.
>
> We would also like to clarify that VIPER _does not_ assume access to ground truth actions, whereas simple behavioral cloning does.
>
> ---
>
> We again thank the reviewer for their insightful comments, and look forward to engaging with them in the reviewer-author discussion period. Please let us know if there are any concerns preventing you from raising your score.
>
> [A] AMP: Adversarial Motion Priors for Stylized Physics-Based Character Control. Peng et al. SIGGRAPH 2021
>
> [B] Generative adversarial imitation from observation. Torabi et al. arXiv 2018
>
> [C] Behavioral Cloning from Observation. Torabi et al. IJCAI 2018

---

> > ### Comment · Reviewer_Zre9 · 2023-08-17
> >
> > I appreciate the authors‘ efforts in answering my questions, but I still have the following concerns.
> >
> > **[Weakness 1 & 2]**
> >
> > I get further confused by the authors' response, which mentions that they trained predictive models on a large dataset, and then used the predictive loss as intrinsic rewards when training (if I understand correctly). So, here comes the first question, the pre-trained dataset may include both go-forward and go-backward tasks, which are the expert behaviors of the two tasks respectively. Then how does the current task know whether to use go-forward or go-backward predictive loss when the downstream task is using the current pre-trained model? Secondly, how can we ensure that the previous (large amount of) expert behaviors are relevant to the expert behavior of the current task? If their distributions are completely uncorrelated, it seems that this predictive loss is not informative at the semantic level, but instead suffers from the problem of overestimation.
> >
> >
> > **[Q2 & Q3]**
> >
> > Why isn't there an overestimation problem, which is common in offline RL? The paper’s use of intrinsic motivation is not fundamentally different from offline RL's use of Q. Intuitively, the overestimation problem exists. Did the authors' experiments not detect this problem? The authors state that "we find that VIPER assigns higher rewards to expert behavior on OOD tasks than to suboptimal and random behaviors on in-distribution tasks". Is it because the model is overfitting? If so, then the intrinsic rewards provided are very sparse.

---

> > > ### Author Response · Authors · 2023-08-18
> > > **Reply to reviewer Zre9**
> > >
> > > We thank the reviewer for engaging and asking clarifying questions. We provide additional clarification below.
> > >
> > > **[Weakness 1 & 2]  Then how does the current task know whether to use go-forward or go-backward predictive loss when the downstream task is using the current pre-trained model?**
> > >
> > > Thank you for pointing this out. In short, we use a **one-hot task label** to modulate the video model for the desired task, when the video model is trained on diverse data.
> > >
> > > We would like to clarify four points:
> > >
> > > * In our experiments, the video model training data always includes the expert target task data, except for cross-embodiment generalization experiments.
> > > * The video model can be trained on the target task-only data (Figure D). This is the single-task imitation learning setting.
> > > * Using multi-task data can further improve the video model with **one-hot task labels**. During RL policy training, we condition the video model on the target task one-hot vector (Figure 3). This is the multi-task imitation learning setting.
> > > * Training the video model with multi-task and multi-agent data enables cross-embodiment generalization (Figure 8 and Figure B). Here, we also use a one-hot task label to specify the target task.
> > >
> > > Sections 4.1 and 4.3 elaborate the training details of our video model but we understand that it is not easy to parse these points. We will revise our paper to make the aforementioned points clear.
> > >
> > > **[Weakness 1 & 2] Secondly, how can we ensure that the previous (large amount of) expert behaviors are relevant to the expert behavior of the current task?**
> > >
> > > Similar to the previous answer, the training dataset includes the expert behavior of the current task, and we specify the target task by feeding a one-hot task label to the video model.
> > >
> > > As can be seen in cross-embodiment experiments (Section 4.3) and OOD generalization analysis (Figure B), using data other than the current task helps generalization capability of our video model.
> > >
> > > **[Q2 & Q3] Why isn't there an overestimation problem, which is common in offline RL?**
> > >
> > > Thank you for raising this question. Offline RL suffers from the overestimation problem mainly due to **bootstrapping** from out-of-distribution actions. Here, only one high value prediction from out-of-distribution actions can be bootstrapped and affect the overall performance significantly.
> > >
> > > As mentioned by Reviewer Zre9, VIPER also has the similar risk that an RL agent exploits out-of-distribution states with high log-probabilities. However, since VIPER measures a probability of a trajectory not a single state, even though one OOD state luckily gets a high reward, it is much less likely to happen throughout the entire trajectory. Thus, our RL agent can learn to follow the distribution of expert trajectories. We will include this discussion in our updated version of the paper.
> > >
> > > **[Q2 & Q3] The authors state that "we find that VIPER assigns higher rewards to expert behavior on OOD tasks than to suboptimal and random behaviors on in-distribution tasks". Is it because the model is overfitting? If so, then the intrinsic rewards provided are very sparse.**
> > >
> > > In Figure B, VIPER assigns higher rewards to expert behaviors on OOD tasks, which haven’t been seen during the video model training (it’s from OOD tasks). Thus, we can conclude that VIPER does not overfit to training data. Instead, it shows a strong generalization capability and thus, provides dense and informative rewards. Additionally, the trained video model achieves a test set loss similar to the train set loss; another indicator that the video model is not overfitting.
> > >
> > > —
> > >
> > > We would like to thank Reviewer Zre9 again for engaging in discussion and providing constructive feedback. We tried our best to address the reviewer’s concerns:
> > > - Conducted additional experiments in Figure D to address the concern “The effectiveness of the proposed method may be limited if the expert data is scarce”. Figure D demonstrates VIPER achieves high performance even with only 10 expert videos (10% of the original dataset size), and learns a less efficient but still meaningful behavior even with only 1 expert video.
> > > - Included additional baselines (GAIfO and BCO), which are shown in Figure A
> > > - Performed additional analysis evaluating the OOD performance of VIPER (Figure B). Notably, we find that VIPER assigns higher rewards to expert behavior on OOD tasks than to suboptimal and random behaviors on in-distribution tasks. The reward histograms show that VIPER reflects different levels of task performance well, even out of distribution.
> > > - Addressed additional concerns regarding VIPER achieving expert-level control and performance compared to baselines (Q4 & Q5).
> > >
> > > Please let us know if you have any concerns and questions!

---

### Author Rebuttal · Authors · 2023-08-10

We thank all reviewers for their constructive feedback and for helping us make our paper a stronger submission!

- First of all, we highlight that we have included **two new imitation learning from observation baselines (Figure A)**, BCO and GAIfO. In short, VIPER consistently shows significant improvement over these baselines on all three task suites. We also hypothesize reasons as to why VIPER outperforms the adversarial and behavioral cloning approaches.

- We have also conducted additional experiments to show the generalization capability of **VIPER on out-of-distribution tasks (Figure B)**. We find that VIPER consistently predicts higher rewards for expert trajectories on both in-distribution and out-of-distribution tasks.

- Additionally, we have added new ablation studies on the use of **suboptimal data (Figure C)**. We found that VIPER can still extract reward signals from suboptimal trajectories. We also observed faster convergence when training from sparse task rewards in conjunction with VIPER rewards trained on suboptimal data.

- Finally, we have added an ablation studying the effect of **dataset size (Figure D)** on the learned VIPER model. We find that VIPER learns a less efficient but still meaningful behavior even with only 1 expert video.

We included these new experimental results in the attached PDF. We hope we have addressed all your concerns and questions. Please let us know if there are any concerns preventing you from raising your score.

---

### Decision · Program_Chairs · 2023-09-21

**Decision:**

Accept (poster)

**Comment:**

The paper proposes novel technique for learning prediction models from unlabeled video and leveraging these models to better enable vision-based behavior learning. How this type of behavior learning should be done is an important question within the NeurIPS community, and the paper provides a compelling and interesting entry in the discussion; it is clear, well-written, and provides compelling empirical evidence that the proposed technique is effective.

The authors successfully engaged in a discussion with the reviewers regarding the ability of the method to generalize in out-of-distribution scenarios, and they’re encouraged to strengthen the original submission with the additional results and discussion on this point and others that arose during the discussion period.